# Unsupervised Noise Adaptive Speech Enhancement by Discriminator-Constrained Optimal Transport

**Hsin-Yi Lin**

The Cooperative Institute for Research in Environmental Sciences (CIRES)
University of Colorado, Boulder, CO, 80309, USA
NOAA Physical Sciences Laboratory, Boulder, CO, 80305, USA
hylin@colorado.edu

**Huan-Hsin Tseng**
Research Center for Information Technology Innovation
Academia Sinica, Taiwan
htseng@citi.sinica.edu.tw

**Xugang Lu**
National Institute of Information and Communications Technology (NICT), Japan
xugang.lu@nict.go.jp

**Yu Tsao**
Research Center for Information Technology Innovation
Academia Sinica, Taiwan
yu.tsao@citi.sinica.edu.tw

## Abstract

This paper presents a novel discriminator-constrained optimal transport network (DOTN) that performs unsupervised domain adaptation for speech enhancement (SE), which is an essential regression task in speech processing. The DOTN aims to estimate clean references of noisy speech in a target domain, by exploiting the knowledge available from the source domain. The domain shift between training and testing data has been reported to be an obstacle to learning problems in diverse fields. Although rich literature exists on unsupervised domain adaptation for classification, the methods proposed, especially in regressions, remain scarce and often depend on additional information regarding the input data. The proposed DOTN approach tactically fuses the optimal transport (OT) theory from mathematical analysis with generative adversarial frameworks, to help evaluate continuous labels in the target domain. The experimental results on two SE tasks demonstrate that by extending the classical OT formulation, our proposed DOTN outperforms previous adversarial domain adaptation frameworks in a purely unsupervised manner.

## 1 Introduction

The goal of speech enhancement (SE) is to convert low-quality speech signals to ones with improved quality and intelligibility. SE serves as an important regression task in the speech-processing field and has been widely used for a pre-processor in speech-related applications, such as speech coding [1], automatic speech recognition (ASR) [2], speaker recognition [3], and assistive hearing devices [4, 5]. Recent advances in machine learning have made significant progress to the SE technology. Generally,

35th Conference on Neural Information Processing Systems (NeurIPS 2021).

learning-based SE approaches estimate a transformation to characterize the mapping function from noisy to clean speech signals in the training phase [6]. The estimated transformation converts noisy speech signals to generate clean-like signals in the testing phase. Various neural network models have been used to characterize noisy-to-clean transformations. Well-known examples of such models include fully connected neural network [7], deep denoising autoencoder [8], convolutional neural network [9], long-short-term memory [10], and Transformer [11]. To effectively handle diverse noisy conditions, we usually prepare a considerable amount of training data that cover various noise types to train SE models. However in real-application scenarios, the noise types in the testing data may not always be involved in the training set. Consequently, the noisy-to-clean transformation learned from the training data cannot be suitably applied to handle the testing noise, resulting in limited enhancement performance. This training-testing mismatch is generally called a domain mismatch issue for SE. An effective solution is required to perform domain adaptation to adjust the SE models with formulating a precise noisy-to-clean transformation that matches the testing conditions. Most existing domain adaptation methods rely on at least one of the following adaptation mechanisms: aligning domain-invariant features [12, 13, 14] and adversarial training, where a discriminator is introduced during training as a domain classifier [15, 16, 17].

This study aims to solve the unsupervised domain adaptation problem for SE by introducing optimal transport (OT). In particular, we consider the circumstance where SE is tested on a target domain with completely unlabeled data, and only labeled data from the source domain is available for reference. Generally speaking, OT theory compares two (probability) distributions and considers all possible transportation plans in between to find one with a minimal displacement cost. The concept of OT can be applied to minimize domain mismatch and consequently achieve unsupervised domain adaptation. Even with the mathematical characteristics offered by OT, obstacles to excellent SE performance persist due to the complex structure possessed by human speech. To further overcome the obstacles, another concept from Generative Adversarial Network (GAN) is integrated to assist attaining sophisticated SE domain adaptation. Although an existing domain transition technique "domain adversarial training" and our proposal share similarity in names, the fundamental constructions are substantially different. A key element in our method lies in a discriminator utilized to examine speech output characteristics, instead of a *domain classifier*. More precisely, the discriminator in our method was employed to govern the output speech quality by learning the probability distribution of the source labels. This novel approach was designed especially for the unsupervised SE domain adaptation to exhibit excellent performance, which was verified on the VoiceBank and TIMIT datasets.

**Contributions**  We proposed a novel method designed specifically for unsupervised domain adaptation in a regression setting. This area of study still has very limited results; moreover, the existing methods often require additional classification of source domains or may not yet be supported by strong regression applications. Conversely, our approach does not require any additional input information other than the source samples, source labels, and target samples. Our approach was applied to two standardized SE tasks, namely VoiceBank-DEMAND and TIMIT, and achieved superior adaptation performance in terms of both Perceptual Evaluation of Speech Quality (PESQ) and Short-Time Objective Intelligibility (STOI) scores. Furthermore, owing to the simple input requirements, we can easily investigate the effect of target sample complexity on our method by increasing the number of noise types allowed in the target domain, which has not been reported by previous literature to the best of our knowledge.

## 2   Related work

**Adversarial domain adaptation:**  The main objective of Domain Adversarial Training (DAT) is to train a deep model (from the source domain) capable of adapting to other similar domains by leveraging a considerable amount of unlabeled data from the target domain [15, 18]. A conventional DAT system consists of three parts, deep feature extractor, label predictor, and domain classifier. By using a gradient reversal layer, the extracted deep features are discriminative for the main learning task and invariant with shifts between the source and target domains. The DAT approach has been applied and confirmed to effectively compensate for the mismatch of source (training time) and target (testing time) conditions in numerous tasks, such as speech signal processing [19, 20], image processing [15, 21], and wearable sensor signal processing [22]. A later development in Multisource Domain Adversarial Networks (MDAN) [23] extended the original DAT to lift the constraint of single

domain transition, utilizing multiple domain classifiers to extract discriminative deep features for the main learning task while being invariant to multiple domain shifts [3, 24].

**Optimal transport for domain adaptation:**   Hitherto, OT  [25, 26] has been utilized to domain adaptation [27, 28] with related analytical results [29]. Furthermore, the concepts of OT have proved itself even more useful under a joint distribution framework [30, 31]. More recently, to improve the sensitivity of OT to outliers, Robust OT was proposed [32]. Moreover, a method for combining the notion of adversarial domain adaptation with OT and margin separation has been proposed [33]. However, almost all experiments performed in these studies were classification problems, unlike the SE task that is the focus of our study.

**Domain adaptation in speech enhancement:**   Existing domain adaptation in SE approaches can be divided into two categories: supervised and unsupervised. For supervised domain adaptation, paired noisy and clean speech signals for the testing conditions are available to adjust the parameters in the SE models. In [34, 35], transfer-learning-based approaches have been proposed to adapt SE models to alleviate corpus mismatches. To combat the catastrophic forgetting issue, Lee et al., proposed a SERIL algorithm that combines curvature-based regularization and path optimization augmenting strategies when preforming domain adaptation on SE models [36]. Conversely, for unsupervised domain adaptation, only noisy speech signals are provided, and the corresponding clean counterparts are not accessible. Generally, unsupervised domain adaptation has good applicability to real-world scenarios. In [37], unsupervised domain adaptation for SE was performed by minimizing the Kullback-Leibler divergence between posterior probabilities produced by teacher and student senone classifiers without paired noisy-clean adaptation data. In [19, 38], the DAT approach was used to adapt SE models to new noisy conditions.

Despite yielding promising performance, the existing unsupervised domain adaptation approaches require additional information, such as word labels, language models, and noise-type labels. In this paper, we propose a new approach: discriminator-constrained OT network (DOTN) to perform unsupervised domain adaptation on SE. In contrast to related works, DOTN does not require additional label information when adapting the original SE models to match new noisy conditions. Our experiments show that DOTN can effectively adapt the SE models to new testing conditions, and that it achieves better adaptation performance than previous adversarial domain adaptation methods, which require additional noise type information.

# 3   Method

## 3.1   Problem setting and notation

Consider a source domain with paired data $(\mathbf{X}^s, \mathbf{Y}^s) = \{(\mathbf{x}_i^s, \mathbf{y}_i^s)\}_{i=1}^{N_s}$, where $\mathbf{x}_i^s \in \mathbb{R}^n$, $\mathbf{y}_i^s \in \mathbb{R}^m$ stand for the input and the corresponding label of sample $i$. The unsupervised domain adaptation assumes that another target domain exists containing only data unlabeled, $\mathbf{X}^t = \{\mathbf{x}_i^t \in \mathbb{R}^n\}_{i=1}^{N_s}$. The goal is to seek a ground truth estimator (or statistical hypothesis) $f : \mathbb{R}^n \to \mathbb{R}^m$ for the target labels $\mathbf{Y}^t = \{\mathbf{y}_i^t\}_{i=1}^{N_t}$ (exist but not known), based on the knowledge provided by the source domain.

The probability distribution of a dataset $\mathcal{D}$ is denoted by $\mathbb{P}_{\mathcal{D}}$, where $\mathcal{D}$ is either $\mathbf{X}^s$, $\mathbf{Y}^s$, $\mathbf{X}^t$ or $\mathbf{Y}^t$ in our discussion. Our problem is to find a function $f$ such that $f$ induces a probability distribution $\mathbb{P}_{f(\mathbf{X}^t)}$ in $\mathbf{Y}^t$ with $\mathbb{P}_{f(\mathbf{X}^t)} \to \mathbb{P}_{\mathbf{Y}^t}$ under certain measure. We propose using the concept of OT to solve this problem.

## 3.2   Proposed model: Discriminator-Constrained Optimal Transport Network (DOTN)

Given a pair of distributions $\mathbb{P}_{\mathcal{D}_1}$ and $\mathbb{P}_{\mathcal{D}_2}$ and a displacement cost matrix $C \geq 0$, OT solves for the transportation plan $\gamma \in \prod(\mathbb{P}_{\mathcal{D}_1}, \mathbb{P}_{\mathcal{D}_2})$ that minimizes the total cost (in a discrete setting)

$$\min_{\gamma \in \prod(\mathbb{P}_{\mathcal{D}_1}, \mathbb{P}_{\mathcal{D}_2})} \langle C, \gamma \rangle_F, \tag{1}$$

where $\prod(\mathbb{P}_{\mathcal{D}_1}, \mathbb{P}_{\mathcal{D}_2})$ denotes the space of joint distributions with marginal $\mathbb{P}_{\mathcal{D}_1}$ and $\mathbb{P}_{\mathcal{D}_1}$. $\langle \cdot, \cdot \rangle_F$ is the Frobenius product, and the entry $C_{ij}$ of $C$ represents the displacement cost of the $i^{th}$ and $j^{th}$

samples. It can be proven that the minimum of this problem is a distance and called the Wasserstein distance when the corresponding cost is a norm [25, 26].

Our proposed method consists of two parts: OT alignment and Wasserstein Generative Adversarial Network (WGAN) training [39, 40]. Both steps are based on OT; however, they are considered with two different pairs of distributions and employ different algorithms.

**Adaptation by Joint Distribution Optimal Transport**    Our adaptation mechanism relies on the alignment between the joint distributions of source and target domains (for the target domain, the label is estimated label). In particular, we approximate $f$ by minimizing the OT loss between the joint distributions $\mathbb{P}_{\mathbf{X}^s} \times \mathbb{P}_{\mathbf{Y}^s}$ and $\mathbb{P}_{\mathbf{X}^t} \times \mathbb{P}_{f(\mathbf{X}^t)}$, with a chosen cost matrix,

$$C_{ij} = \alpha \left\| \mathbf{x}_i^s - \mathbf{x}_j^t \right\|^2 + \beta \left\| \mathbf{y}_i^s - f(\mathbf{x}_j^t) \right\|^2, \qquad (\alpha, \beta > 0) \tag{2}$$

By aligning the joint distributions of the source and target domains, noise adaptation is naturally achieved as OT seeks the source sample that is the most "similar" for each target sample.

Although the OT provides accurate estimates for each sample, the effect of each estimation error could accumulate in the training process and mislead $f$ to a convenient local minimum without preserving the speech data structure. To avoid this situation, we employ Wasserstein Generative Adversarial Network (WGAN) training to complement and enhance our adaptation system.

**Discriminative training on outputs and source labels**    Distinct from the adaptation where we consider the joint distributions of the inputs and labels, we focus on WGAN training for the source label distribution $\mathbb{P}_{\mathbf{Y}^s}$ and output distribution $\mathbb{P}_{f(\mathbf{X}^t)}$. In the terminology of generative adversarial training, we consider $f$ a generator and introduce a convolutional neural network-based discriminator, $h$, as the 'critic'. In general, we use the discriminator to decide whether the outputs of $f$ are 'similar' to the source labels. Formally, the WGAN algorithm solves

$$\min_f \max_{h \in \mathcal{L}} \left\{ \mathbb{E}_{y \sim \mathbb{P}_{\mathbf{Y}^s}} (h(y)) - \mathbb{E}_{x \sim \mathbb{P}_{\mathbf{X}^t}} (h(f(x))) \right\} \tag{3}$$

by Kantorovich-Rubinstein duality [25], where $\mathcal{L}$ is the set of 1-Lipschitz functions. In this case, under an optimal discriminator minimizing the value function with respect to the generator parameters minimizes the Wasserstein distance between the distributions $\mathbb{P}_{\mathbf{Y}^s}$ and $\mathbb{P}_{f(\mathbf{X}^t)}$.

This discriminative training complements our alignment and provides additional constraints from the explicit relation between the source labels and estimations of target labels. These constraints support the joint distribution alignment and provides further guidance in the gradient descent training process. The performance of the experiments is considerably improved when our discriminative training supports the joint distribution alignment.

### 3.3   Loss functions and the proposed algorithm

Our domain alignment can be achieved by solving the following optimization problem:

$$\min_{\gamma, f} \mathcal{L}_1 + \mathcal{L}_2 = \min_{\gamma, f} \frac{1}{N^s} \sum_i \left\| \mathbf{y}_i^s - f(\mathbf{x}_i^s) \right\|^2 + \sum_{i,j} \gamma_{ij} \left( \alpha \left\| \mathbf{x}_i^s - \mathbf{x}_j^t \right\|^2 + \beta \left\| \mathbf{y}_i^s - f(\mathbf{x}_j^t) \right\|^2 \right), \tag{4}$$

where $\alpha, \beta > 0$ are the parameters chosen for balance. Notably, the first term emphasizes the knowledge from the source domain is not to be forgotten during training, which has been revealed in several works [31, 41, 42]. Without this emphasis, the source domain knowledge cannot be well-maintained, and thus the overall performance may degrade. Such phenomenon was also observed in the SE experiments. The second term is intended for domain alignment. To show some intuitions, consider the ideal case where Eq. (4) is completely minimized to zero, which leads to

$$\left\| \mathbf{x}_i^s - \mathbf{x}_j^t \right\|^2 \equiv 0 \text{ and } \left\| \mathbf{y}_i^s - f(\mathbf{x}_j^t) \right\|^2 \equiv 0 \quad \Rightarrow \quad \mathbf{x}_i^s = \mathbf{x}_j^t \text{ and } \mathbf{y}_i^s = f(\mathbf{x}_j^t) \tag{5}$$

for all $i, j$. This indicates that for each given target domain sample $\mathbf{x}_j^t$, an identical sample $\mathbf{x}_i^s$ from the source domain is found, and the unknown target label is then constructed by the corresponding source label. Even though practically the ideal case of zero loss is unlikely to happen, the OT loss aims to search for the most "similar" correspondence, which entails the intuition of domain alignment

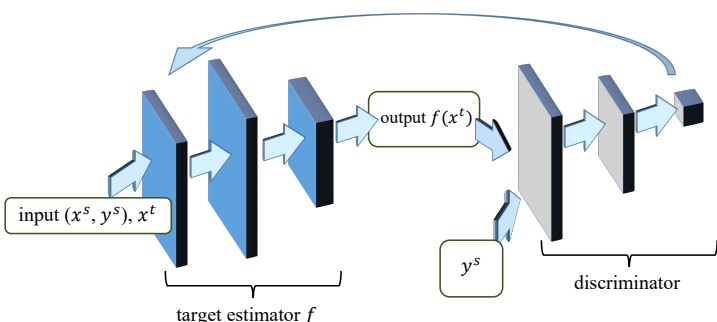

Figure 1: DOTN network structure.

for Eq. (4). From this point of view, it is noted that although the term $\|\mathbf{x}_i^s - \mathbf{x}_j^t\|$ in $C_{ij}$ (in Eq. (2)) is not directly related to the network backpropagations of $f$ and $h$, it may not be ignored as the discard of this term will result in a wrong transportation plan $\gamma_{ij}$ and eventually lead to an undesired alignment.

For discriminative training, the discriminator $h$ is trained by the discriminator loss function $\mathcal{L}_h = \frac{1}{m}\sum_{i=1}^m h(\mathbf{y}_i^s) - h(f(\mathbf{x}_i^t))$, and $f$ follows the generator loss function $\mathcal{L}_f = -\frac{1}{m}\sum_{i=1}^m h(f(\mathbf{x}_i^t))$ where $m$ is the batch size. As there are multiple sets of parameters $\gamma$, $h$ and $f$ in our framework, one set of parameters is updated each time, while the other sets of parameters are fixed.

---

**Algorithm 1** DOTN, proposed algorithm

---

**Require:** $\mathbf{x}^s$, source domain inputs. $\mathbf{y}^s$, source domain labels. $\mathbf{x}^t$, target domain inputs. $c$, the clipping parameter. $m$, the batch size. $n_f, n_h, n_s$: number of iterations of OT per generator training, discriminator training, and source domain training, respectively. $n$, number of iterations.
**Require:** $\theta_f$, initial parameters of estimator $f$. $\theta_h$, initial parameters of discriminator $h$.
1: **for** each batch of source samples $(\mathbf{x}^s, \mathbf{y}^s)$ and target samples $(\mathbf{y}^t)$ **do**
2:     fix $\theta_f$, solve for $\gamma$ in Eq. (4) by OT.
3:     fix $\gamma$, $\theta_f \leftarrow \text{Adam}(\nabla_{\theta_f}\mathcal{L}_2, \theta_f, \theta_h)$.
4:     **if** $n \bmod n_s == 0$ **then**
5:         $\theta_f \leftarrow \text{Adam}(\nabla_{\theta_f}\mathcal{L}_1, \theta_f, \theta_h)$.
6:     **end if**
7:     **if** $n \bmod n_f == 0$ **then**
8:         $\theta_f \leftarrow \text{Adam}(\nabla_{\theta_f}\mathcal{L}_f, \theta_f, \theta_h)$.
9:     **end if**
10:    **if** $n \bmod n_h == 0$ **then**
11:        $\theta_h \leftarrow \text{Adam}(\nabla_{\theta_h}\mathcal{L}_h, \theta_f, \theta_h)$.
12:        $\theta_h \leftarrow \text{clip}(\theta_h, -c, c)$.
13:    **end if**
14: **end for**

---

## 4 Experiments

We evaluated our method in SE on two datasets: Voice Bank corpus [43] and TIMIT [44]. Although there is abundant literature on unsupervised domain adaptation, very limited methods have been successfully applied to SE or regression problems. Closely relevant methods often require additional input structures or domain label. Unlike previous methods, our approach does **not** rely on additional data information. Nevertheless, we compared our results with two most relevant adversarial domain adaptation methods: DAT [19] and MDAN [23]. The original MDAN was designed for classification problems and cannot be directly applied on SE. Preserving the fundamental ideas, we modified the MDAN structure for regressions, so that SE experiments can be performed for comparison. The implementations are summarized in the supplementary material and codes are available on Github[1].

---

[1]https://github.com/hsinyilin19/Discriminator-Constrained-Optimal-Transport-Network

## 4.1 Comparisons to DAT and MDAN

As both DAT[2] and MDAN[3] utilize a domain classifier to derive domain invariant features, domain labels become essential in their adaptation mechanism. Thus, both these methods are considered *weakly supervised*. Specifically, they request inputs of the form $\{\mathbf{x}_i^s, \mathbf{y}_i^s, c_i\}_i$ and $\{\mathbf{x}_j^t, c_j\}_j$ with $c_i, c_j \in \mathcal{K}$, where $\mathcal{K}$ is an index set specifying the origin of data domains. For DAT, $\mathcal{K} = \{0, 1\}$ to indicate whether a sample belongs to the source or target domain, while for MDAN $\mathcal{K} = \{0, 1, \ldots, K\}$ to indicate that there are $K$ distinct source domains and one target domain. The requirement on domain labels poses restrictions in certain circumstances, such as a new target sample may not always fall into any existing categories, or the data origin is simply unknown. In contrast, the proposed method DOTN is not bounded by specifications of data origin and simply receives inputs of the form $\left\{\mathbf{x}_i^s, \mathbf{y}_i^s, \mathbf{x}_j^t\right\}_{i,j}$. With less input requirement, DOTN is more flexible to be applied in various scenarios, more approachable to real-world applications.

## 4.2 Voice Bank with DEMAND noise database

**Dataset**  In the first set of experiments, the pre-selected subset of Voice Bank provided by [45] was used to test the proposed DOTN. For source domain data, 14 male speakers and 14 female speakers were randomly selected out of totally 84 speakers (42 male and 42 female), and each speaker pronounced around 400 sentences. As a result, the clean data in source domain contained 5,724 male utterances and 5,848 female utterances, amounting to 11,572 utterances in total. We then mixed the 11,572 clean utterances with noise from DEMAND [46], in transportation category: "Bus", "Car", "Metro" at 7 Signal-to-noise ratio (SNR) levels (-9, -6, -3, 0, 3, 6, and 9 dB). Accordingly, for this set of source domain data, both noisy speech signals and the corresponding clean references are prepared.

A target domain data contained $5,768$ noisy utterances mixed by 824 clean ones from two random speakers (1 male, 1 female) (followed the design in [47]) with one of the three noise types from DEMAND, in STREET category: "Traffic", "Cafe", or "Public square" and 7 SNR ratios (-9, -6, -3, 0, 3, 6, and 9 dB). No clean labels were given under target domain. That is, for the target domain data, only noisy speech signals are provided and the corresponding clean references are not accessible.

All source samples (both noisy utterances and the corresponding clean references, prepared from the three transportation noise types) and the target samples (only noisy utterances without the corresponding clean references, prepared from the one street noise type) were included in our training set. We conducted the experiments under single-target noise-type circumstances and compared the results with DAT and MDAN. More specifically, we ran this setting for all three cases, where the target domain contained either cafe, public square, or traffic noise.

**Results**  Table 1 and Table 2 list the PESQ and STOI scores, respectively, of DAT, MDAN and DOTN under three noise types at 7 SNR levels. "Avg" denotes the averaged scores over 7 SNR levels. From the PESQ scores reported in Table 1, we note that the proposed DOTN outperforms both DAT and MDAN consistently over different noise types and SNR levels, except for the Cafe noise type at -9 dB SNR. This might be owning to a potential limitation of DOTN, which will be detailed in the next section. Next, from Table 2, we note that STOI scores show very similar trends to that of PESQ scores, as listed in Table 1.

---

[2]DAT consists of three components, a deep feature extractor $\mathcal{E} : \mathcal{X} \to \mathcal{Z}$, a (task) label predictor $F_Y : \mathcal{Z} \to \mathcal{Y}$, and a domain classifier $F_D : \mathcal{Z} \to \mathcal{K}$, to form two functional pairs: $F_D \circ \mathcal{E}$ and $F_Y \circ \mathcal{E}$. Here $\mathcal{X}, \mathcal{Y}$ are the input space and (task) label space respectively. $\mathcal{Z}$ denotes the invariant (latent) feature space; $\mathcal{K} = \{0 : \text{source}, 1 : \text{target}\}$ as the domain label classes. The pair $F_D \circ \mathcal{E} : \mathcal{X} \to \mathcal{K}$ formed by $F_D, \mathcal{E}$ works against each other, by a Gradient Reversal Layer, to derive domain invariant features in $\mathcal{Z}$. Another pair $F_Y \circ \mathcal{E} : \mathcal{X} \to \mathcal{Y}$ demands that the invariant features in $\mathcal{Z}$ encode sufficient information for (main task) label classifications at the same time. Via adversarial training, the two pairs eventually reach a balance completing the main learning task as well as eliminating the domain mismatch.

[3]MDAN inherits the DAT architecture with an extension to $K$ source domains and $K$ domain classifiers $F_{D_i} : \mathcal{Z} \to \{0, 1\}, i = 1, \ldots K$ with similar adversarial training applied.

Table 1: PESQ scores for VoiceBank-DEMAND

| noise type | Traffic | | | Cafe | | | Public square | | |
|---|---|---|---|---|---|---|---|---|---|
| SNR(dB)/model | DAT [19] | MDAN [23] | DOTN | DAT | MDAN | DOTN | DAT | MDAN | DOTN |
| -9 | 1.307 | 1.670 | **1.863** | 1.058 | **1.539** | 1.497 | 1.436 | 1.929 | **2.037** |
| -6 | 1.446 | 1.920 | **2.182** | 1.184 | 1.735 | **1.853** | 1.655 | 2.119 | **2.258** |
| -3 | 1.718 | 2.153 | **2.395** | 1.362 | 1.949 | **2.089** | 1.939 | 2.318 | **2.439** |
| 0 | 2.081 | 2.366 | **2.591** | 1.599 | 2.153 | **2.332** | 2.268 | 2.512 | **2.601** |
| 3 | 2.381 | 2.535 | **2.740** | 1.865 | 2.345 | **2.490** | 2.575 | 2.670 | **2.761** |
| 6 | 2.712 | 2.708 | **2.888** | 2.189 | 2.534 | **2.661** | 2.867 | 2.824 | **2.889** |
| 9 | 3.016 | 2.854 | **3.015** | 2.493 | 2.695 | **2.783** | 3.120 | 2.966 | **3.057** |
| Avg | 2.094 | 2.315 | **2.525** | 1.679 | 2.136 | **2.244** | 2.266 | 2.477 | **2.577** |

Table 2: STOI scores for VoiceBank-DEMAND

| noise type | Traffic | | | Cafe | | | Public square | | |
|---|---|---|---|---|---|---|---|---|---|
| SNR(dB)/model | DAT [19] | MDAN [23] | DOTN | DAT | MDAN | DOTN | DAT | MDAN | DOTN |
| -9 | 0.584 | 0.708 | **0.721** | 0.557 | **0.643** | 0.633 | 0.667 | 0.747 | **0.765** |
| -6 | 0.659 | 0.761 | **0.790** | 0.616 | 0.702 | **0.723** | 0.725 | 0.792 | **0.815** |
| -3 | 0.728 | 0.810 | **0.833** | 0.687 | 0.760 | **0.779** | 0.771 | 0.830 | **0.850** |
| 0 | 0.785 | 0.849 | **0.871** | 0.741 | 0.805 | **0.831** | 0.815 | 0.859 | **0.877** |
| 3 | 0.824 | 0.874 | **0.894** | 0.785 | 0.842 | **0.861** | 0.845 | 0.881 | **0.900** |
| 6 | 0.860 | 0.896 | **0.914** | 0.825 | 0.871 | **0.887** | 0.872 | 0.898 | **0.915** |
| 9 | 0.885 | 0.910 | **0.927** | 0.854 | 0.893 | **0.905** | 0.893 | 0.914 | **0.931** |
| Avg | 0.761 | 0.830 | **0.850** | 0.724 | 0.788 | **0.803** | 0.798 | 0.846 | **0.865** |

## 4.3 TIMIT

**Dataset** For the second part of experiments, TIMIT was used to prepare the source and target samples. The *clean speech* of the source domain $\{\mathbf{y}_i^s\}_{i=1}^{N_s}$ for training consists of utterances, contributed by 48 male speakers and 24 female speakers from 8 dialect regions. Each speaker had 8 sentences, including 5 SX (phonetically compact sentences) and 3 SI (phonetically diverse sentences), according to the official suggestion of TIMIT. The number of the speakers selected was to maintain the balance of the original data.

The above clean utterances were used to mix with 5 *stationary* noise types (Car, Engine, Pink, Wind, and Cabin) at 9 SNR levels (-12, -9, -6, -3, 0, 3, 6, 9, and 12 dB), amounting to 25,920 noisy utterances, to be the *noisy speech* of the source domain $\{\mathbf{x}_i^s\}_{i=1}^{N_s}$ with $N_s = 25,920$.

For the target domain, a total of 24 speakers suggested by TIMIT core test set was all used to have 192 clean utterances for the target domain $\mathbf{y}_i^t$, which were subsequently mingled with one of the four *non-stationary* noise types: "Helicopter", "Cafeteria", "Baby-cry", or "Crowd-party" under 7 SNRs (-9, -6, -3, 0, 3, 6, and 9 dB) as target inputs $\{\mathbf{x}_i^t\}_{i=1}^{N_t}$ with $N_t = 1,344$. The choice of noise types for the source and target domain was to let the learning algorithms adapt from distinguished environments in the real-world.

**Results** Table 3 lists the PESQ and STOI scores of the DAT, MDAN, and DOTN under four noise types at seven SNR levels. From the table, we first note that DOTN consistently outperforms DAT and MDAN in terms of both "Avg" PESQ and STOI scores among the four noise types. With a more careful investigation, the DOTN achieves higher PESQ and STOI scores over DAT and MDAN for all SNR conditions in the Helicopter noise. For the Crowd-party and Cafeteria noises, DOTN outperforms DAT and MDAN in most higher SNR conditions for both PESQ and STOI scores. However, for the Baby-cry noise, DOTN outperforms DAT and MDAN in STOI but underperforms MDAN in PESQ. Note that Cafeteria, Crowd-party, and Baby-cry noise types involved clear human speech components, which may cause confusions when DOTN tries to retrieve the target speech (clean reference). Thus, DOTN yields sub-optimal performance when dealing with these noise types, especially under very low SNR conditions, where background speech components might overwhelm the target speech. Nevertheless, the overall average PESQ and STOI scores of DOTN (P=1.838; Q=0.7238) are still higher than that of DAT (P=1.5728; Q=0.6298) and MDAN (P=1.8001; Q=0.7038), where P and Q denote the PESQ and STOI scores, respectively, over the 4 noise types and

Table 3: TIMIT results

| noise type | Helicopter | | | | | | Crowd-party | | | | | |
|---|---|---|---|---|---|---|---|---|---|---|---|---|
| model | DAT [19] | | MDAN [23] | | DOTN | | DAT | | MDAN | | DOTN | |
| SNR(dB) | PESQ | STOI | PESQ | STOI | PESQ | STOI | PESQ | STOI | PESQ | STOI | PESQ | STOI |
| -9 | 1.031 | 0.392 | 1.252 | 0.517 | **1.455** | **0.577** | **1.483** | **0.544** | 1.150 | 0.440 | 1.056 | 0.451 |
| -6 | 1.015 | 0.431 | 1.443 | 0.594 | **1.669** | **0.649** | **1.484** | **0.560** | 1.356 | 0.516 | 1.302 | 0.538 |
| -3 | 1.094 | 0.497 | 1.664 | 0.670 | **1.890** | **0.716** | 1.528 | 0.592 | **1.560** | 0.600 | 1.559 | **0.621** |
| 0 | 1.268 | 0.566 | 1.902 | 0.742 | **2.104** | **0.775** | 1.596 | 0.636 | 1.776 | 0.684 | **1.816** | **0.709** |
| 3 | 1.518 | 0.637 | 2.134 | 0.801 | **2.289** | **0.822** | 1.736 | 0.690 | 1.986 | 0.762 | **2.042** | **0.782** |
| 6 | 1.779 | 0.701 | 2.363 | 0.849 | **2.497** | **0.865** | 1.953 | 0.750 | 2.179 | 0.823 | **2.236** | **0.838** |
| 9 | 2.094 | 0.759 | 2.563 | 0.884 | **2.677** | **0.895** | 2.200 | 0.801 | 2.355 | 0.865 | **2.447** | **0.885** |
| Avg | 1.400 | 0.569 | 1.903 | 0.722 | **2.083** | **0.757** | 1.711 | 0.653 | 1.766 | 0.670 | **1.780** | **0.690** |

| noise type | Cafeteria | | | | | | Baby-cry | | | | | |
|---|---|---|---|---|---|---|---|---|---|---|---|---|
| model | DAT | | MDAN | | DOTN | | DAT | | MDAN | | DOTN | |
| SNR(dB) | PESQ | STOI | PESQ | STOI | PESQ | STOI | PESQ | STOI | PESQ | STOI | PESQ | STOI |
| -9 | 1.196 | 0.440 | **1.248** | **0.471** | 1.185 | 0.436 | 1.109 | 0.580 | **1.116** | 0.561 | 0.984 | **0.597** |
| -6 | 1.206 | 0.471 | 1.395 | **0.535** | 1.419 | 0.528 | 1.313 | 0.630 | **1.313** | 0.636 | 1.180 | **0.668** |
| -3 | 1.244 | 0.519 | 1.608 | 0.614 | **1.631** | **0.620** | 1.495 | 0.665 | **1.500** | 0.699 | 1.431 | **0.732** |
| 0 | 1.408 | 0.579 | 1.827 | 0.692 | **1.859** | **0.699** | 1.621 | 0.707 | **1.734** | 0.768 | 1.665 | **0.789** |
| 3 | 1.631 | 0.651 | 2.031 | 0.764 | **2.075** | **0.769** | 1.801 | 0.746 | **1.909** | 0.809 | 1.889 | **0.835** |
| 6 | 1.915 | 0.719 | 2.244 | 0.822 | **2.271** | **0.826** | 1.973 | 0.782 | **2.112** | 0.849 | 2.105 | **0.871** |
| 9 | 2.216 | 0.782 | 2.422 | 0.864 | **2.458** | **0.873** | 2.133 | 0.814 | 2.274 | 0.880 | **2.277** | **0.890** |
| Avg | 1.545 | 0.594 | 1.825 | **0.680** | **1.843** | 0.679 | 1.635 | 0.703 | **1.708** | 0.743 | 1.647 | **0.769** |

9 SNR levels. Please also note that DAT and MDAN require additional domain label information [19] while DOTN performs domain adaptation in a purely unsupervised manner.

To show the advantages and flexibility of DOTN, we further explored and applied DOTN to the circumstances when multiple noise types are included in the target domain. Fig 2 and Fig 3, respectively, present the PESQ and STOI results over four SNR levels on multiple target noise types; here H, Ca, Cr, and B denote Helicopter, Cafeteria, Crowd-party, and Baby-cry noise types, respectively. In Fig 2, when the testing noise is "Helicopter", the PESQ scores of using two noise types (H+Ca, H+Cr, H+B) and three noise types (H+Cr+B, H+Cr+Ca, H+B+Ca) are comparable. Further, the PESQ scores of these six systems are also comparable to that of using one noise type (H). The results confirmed that the DOTN has minor effects on catastrophic forgetting, which is a common issue in domain adaptation approaches [48, 49]. In the case of 4 noise types (the system sequentially learned Helicopter, Baby-cry, Cafeteria, and Crowd-party noise types and was tested on the Helicopter noise), the achieved performance drops moderately. We observe very similar trends for the other three noise types (baby-cry, cafeteria, and crowd-party) in Fig 2.

From Fig 3, similar trends as those from Fig 2 are observed: (1) The seven results using 1, 2, and 3 noise types are comparable, thereby confirming that the DOTN has minor effects on catastrophic forgetting. (2) When there are 4 noise types for sequential learning, the achievable STOI scores start to decrease.

## 5   Conclusion

In this study, we proposed a novel DOTN method, which was designed specifically for unsupervised domain adaptation in regression setting. Our approach skillfully fuses OT and generative adversarial frameworks to achieve unsupervised learning in target domain based on the information provided from the source domain, requiring no additional structure or inputs, such as multi-source domain and noise type labels. Our experiments show that the proposed method is capable of superior adaptation performance in SE by outperforming other adversarial domain adaptation methods on both PESQ and STOI scores for the VoiceBank-DEMAND and TIMIT datasets. Further, we show that when moderately increasing the complexity of the target samples (by increasing the number of noise types in the target domain), only a small degree of degradation was observed. This suggests that our method is robust to sample complexity in the target domain.

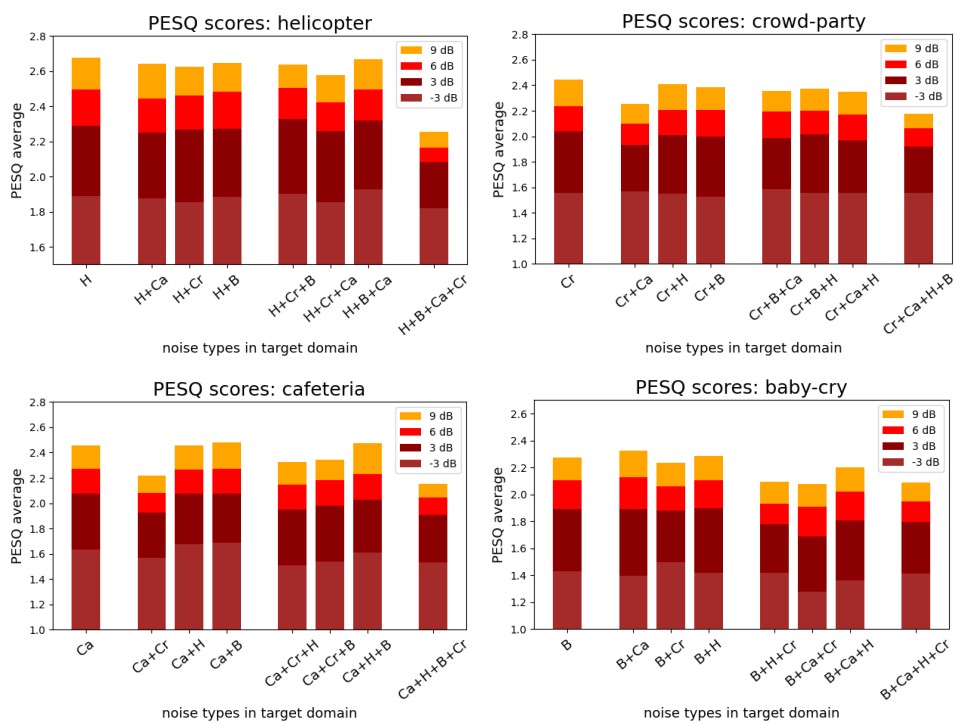

Figure 2: Comparisons in PESQ average for cases with multiple noise types in target domain, where H, Ca, Cr, and B denote Helicopter, Cafeteria, Crowd-party, and Baby-cry noise types.

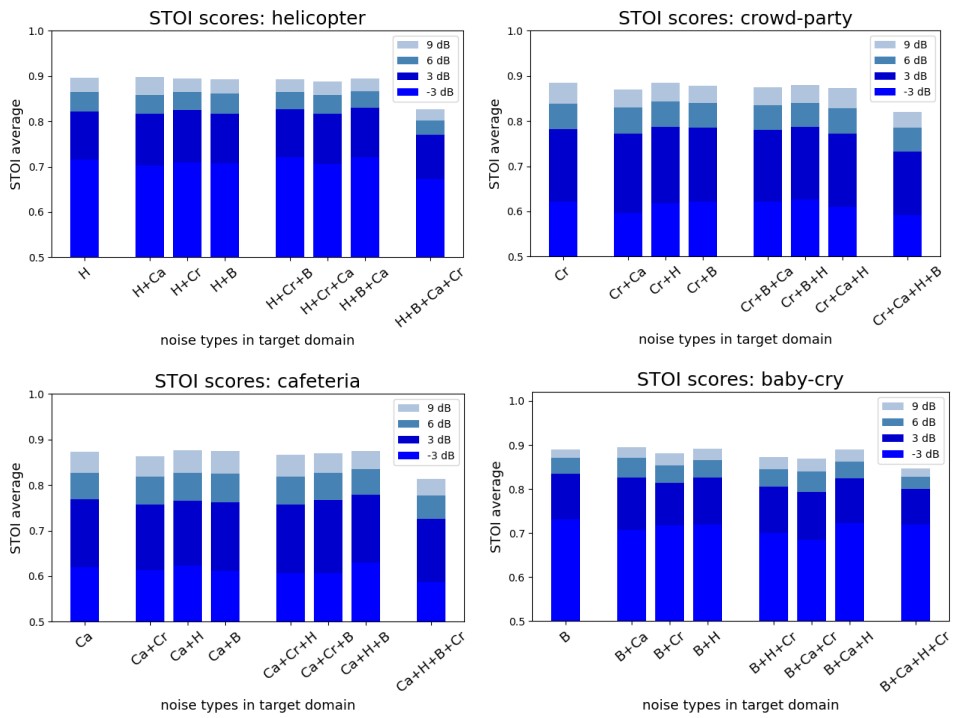

Figure 3: Comparisons in STOI average for cases with multiple noise types in target domain.

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
