# Discriminator-Constrained Optimal Transport for Unsupervised Noise Adaptive Speech Enhancement –Supplementary Material–

**Hsin-Yi Lin**
The Cooperative Institute for Research in Environmental Sciences (CIRES)
University of Colorado, Boulder, CO, 80309, USA
NOAA Physical Sciences Laboratory, Boulder, CO, 80305, USA
hylin@colorado.edu

**Huan-Hsin Tseng**
Research Center for Information Technology Innovation
Academia Sinica, Taiwan
htseng@citi.sinica.edu.tw

**Xugang Lu**
National Institute of Information and Communications Technology (NICT), Japan
xugang.lu@nict.go.jp

**Yu Tsao**
Research Center for Information Technology Innovation
Academia Sinica, Taiwan
yu.tsao@citi.sinica.edu.tw

In this supplementary material, we provide implementation details, audio demonstrations, enhancement quality evaluations, as well as spectrograms and waveforms of some experimental results. The corresponding audio files of the presented results are accessible at `https://drive.google.com/drive/folders/1fuZIqM-feg4CUnU-zNeUCl_6IOARJOPg?usp=sharing` and the codes of proposed method in the Github repository: `https://github.com/hsinyilin19/Discriminator-Constrained-Optimal-Transport-Network`.

## 1 Implementation details

**Corpus** We used Voice Bank (VCTK) [1] and TIMIT datasets available online. VCTK can be downloaded at https://datashare.is.ed.ac.uk/handle/10283/3443 and TIMIT can be found at: https://catalog.ldc.upenn.edu/LDC93S1.

**Noise database** The environmental noise recordings- DEMAND [2] mixed with Voice Bank in the experiments can be downloaded at: `https://doi.org/10.5281/zenodo.1227121`. The five stationary noises (Car, Engine, Pink, Wind, and Cabin) and four nonstationary noises (Helicopter, Cafeteria, Baby-cry, and Crowd-party) used in the TIMIT experiment can be found in our Github repository.

**Data processing** All corpora are in the WAV format with a 16 kHz sampling rate. Data preprocessing code is provided to generate clean speech from scratch for training and testing stage. Additionally,

35th Conference on Neural Information Processing Systems (NeurIPS 2021).

another code mixing selected types of noise at a variety of SNR levels to clean utterances is provided. For computational convenience, waveforms were converted into STFT spectrograms.

**Network structures**

- The DAT [3] model was based on the optimal architecture provided on Github: `https://github.com/jerrygood0703/noise_adaptive_DAT_SE`, where two consecutive Bi-directional Long Short-Term Memory (BiLSTM) of 512 hidden units were used to connect with one fully-connected-layer of 1024 nodes for the SE generator. The domain classifier consisted of one LSTM of 1024 hidden units to connect with a fully-connected-layer of 1024 nodes for binary classification.

- Since MDAN [4] was not designed for regression tasks in the first place, much modification was required to fit the SE purpose. The original design used $k$ label (task) classifiers as well as $k$ domain classifiers for each of $k$ source domains. First, an encoder consisted of one BiLSTM of 512 hidden units was used to encode the input (spectrum) into 512-dim domain-insensitive latent variables. Subsequently, the original $k$ label classifiers were replaced by $k$ SE generators for output dimension 257, each of which contained one BiLSTM of 512 hidden units and a fully-connected-layer of 1024 nodes for regression outputs. The original MDAN code of classifications is provided: `https://github.com/hanzhaoml/MDAN.git`; our modification for speech enhancement is `https://github.com/hsinyilin19/Discriminator-Constrained-Optimal-Transport-Network`.

  In the meantime, there were $k$ additional source domain classifiers attempting to produce 512-dim domain-insensitive latent variables by applying the technique of Gradient Reversal Layers; each source domain classifier was comprised of one LSTM of 1024 hidden nodes and one fully-connected-layer of 1024 nodes for final binary classification.

- For DOTN, the discriminator was comprised of two consecutive 2D-Convolutional Neural Network (CNN) of kernel size 5 and subsequently two fully-connected-layers (16384 nodes and 256 nodes) to discriminate signals from the generator or not (True/False), where ReLu was used in between layers and Sigmoid for the final output. The generator was composed of a 2-layer BiLSTM of 512 hidden units to connect with two fully-connected-layers of 1024 nodes and 512 nodes, respectively.

**Optimization and hyperparameters**

- DAT was based on Tensorflow 1.6, where the ADAM optimizer with learning rate $10^{-4}$ and batch size 16 was adopted to train the model with $10^5$ iterations for TIMIT and $5 \times 10^4$ iterations for VCTK, respectively.

- MDAN used the ADAM optimizer with learning rate $10^{-3}$ and batch size 1800 to train 60 epochs for TIMIT and 5 epochs for VoiceBank-DEMAND, respectively. The ratio between the two losses of SE generator and domain classifier is 0.001.

- DOTN used the ADAM optimizer for both discriminator and generator with batch size 1800 to train 10 epochs for VoiceBank-DEMAND and 60 epochs for TIMIT. There were several training steps in the proposed method, each was set at a different learning rate. The OT alignment was trained with learning rate $10^{-5}$, the source domain knowledge with $10^{-4}$, the generator training with $10^{-5}$, and discriminator training with $10^{-3}$. $\alpha$ and $\beta$ were both fixed at 1, and the clipping parameter for discriminator was set at $10^{-3}$.

  To balance OT and adversarial training, we used different training frequency for each part of the proposed method to reach different levels of control strengths. From our experience, a successful training commonly happens when the training frequency is set from high to low in the following order: OT alignment, discriminator training, and then generator training. For example, the discriminator training was performed once every 5 iterations of OT alignment, and generator training was once every 10 iterations of OT alignment for TIMIT. On VoiceBank-DEMAND, the OT alignment and discriminator training had the same frequency, but the generator training was performed once every 2 iterations of OT alignment.

**Hardware**  All experiments were run on one NVIDIA Tesla V100 GPU of 32 GB CUDA memory and 4 CPUs with 90 GB memory.

**Runtime**

- (DAT) Around 4 hours training time on TIMIT ($10^5$ training iterations), and 3 hours training time on VoiceBank-DEMAND ($5 \times 10^4$ training iterations)

- (MDAN) Approximately 4 hours on TIMIT (60 epochs), and approximately 30 hours on VoiceBank-DEMAND (5 epochs).

- (DOTN) It consumed approximately 5 hours for each DOTN trial on TIMIT (60 epochs), and approximately 15 hours on VoiceBank-DEMAND (10 epochs).

## 2 Additional experimental results

### 2.1 Visualization of SE outputs

In the main manuscript, we present and discuss the quantitative results (in-terms of PESQ and STOI scores) of the proposed DOTN and compared methods, namely, DAT and MDAN. In this supplementary file, we present the waveform and spectrogram plots of the enhanced utterances produced by MDAN, DAT, and DOTN for qualitative analyses. A spectrogram plot is a popular tool to analyze the time-frequency characteristics of speech signals [5]. In Figs. 1 and 2, respectively, we demonstrate the waveform and spectrogram plots for an utterance pronounced by a male speaker from Voice Bank (no.232) contaminated with Cafe background noise from DEMAND at 0dB SNR level; the corresponding clean reference and enhanced versions by MDAN, DAT, and DOTN are also presented. In both figures, the top panels present the noisy utterance (right) and its clean version (left). The bottom panels demonstrate the enhanced results provided by MDAN (left), DAT (middle), and DOTN (right). From Figs. 1 and 2, MDAN, DAT, and DOTN all successfully suppress noise components given the noisy utterance. Among them, DAT seems to yield the best noise suppression result. With a further investigation on the spectrogram plot (Fig. 2), however, we note that some detailed speech structures of the DAT output are distorted, and some speech components are removed, as marked by yellow rectangular regions. The results from Figs. 1 and 2 clearly show that the qualitative results are consistent with those of the quantitative results (PESQ and STOI scores) as reported in the main manuscript. Next, Figs. 3 and 4, respectively, demonstrate the waveform and spectrogram plots of an utterance pronounced by a female speaker in Voice Bank (no.257) contaminated with the Cafe background at 0 dB SNR level along with its clean reference and enhanced versions. From Figs. 3 and 4, we note the same trends as those from Figs. 1 and 2: (1) MDAN, DAT, and DOTN all effectively suppress noise components given the noisy input, and DAT seems to yield the best noise suppression result. (2) As compared to DAT, DOTN can more effectively preserve detailed speech structures, as marked by the yellow rectangular regions in Fig. 4.

We further drew the waveform and spectrogram plots of utterances pronounced by one male and one female speaker from the TIMIT dataset. Figs. 5 and 6, respectively, are the waveform and spectrogram plots for a male speaker (labeled MTLS0), and Figs. 7 and 8, respectively, are the waveform and spectrogram plots for a female speaker (labeled FDHC0); both utterances were contaminated with Cafeteria background at 0dB SNR level; the clean references and enhanced versions are also presented in the figures. The qualitative results of the TIMIT dataset show very similar trends to that of "VoiceBank+DEMAND" (from Figs. 1 to 4). All of the three methods MDAN, DAT, and DOTN can suppress noise components while DOTN provides better results than the other two methods. The advantages of DOTN over DAT are marked by yellow rectangular regions in the spectrogram plots (Figs. 6 and 8). In summary, from Figs. 1 to 8, we note that the qualitative results align well with the quantitative scores as reported in the main manuscript. Please also note that our listening tests indicate that enhanced utterances by DOTN yields better quality with less distortion effects as compared to MDAN and DAT. Please refer to our audio samples: `https://drive.google.com/drive/folders/1fuZIqM-feg4CUnU-zNeUCl_6IOARJOPg?usp=sharing`.

Finally, we would like to make remarks on the structure of the proposed method DOTN. While the mechanism for domain adaptation of DOTN relies mainly on the joint distribution OT, the adversarial training is crucial for the enhanced speech quality. In fact, the MSE loss of spectrum was involved in our OT alignment, which (if used solely) could result in 'impetuous' erasing effect on the speech data and low speech quality as we observed in experiments. This phenomenon was also observed in the case of DAT [3], also a MSE-based method. The introduction of discriminator is our solution for attacking this issue. Based on the clean utterances in source domain (as references), the discriminator

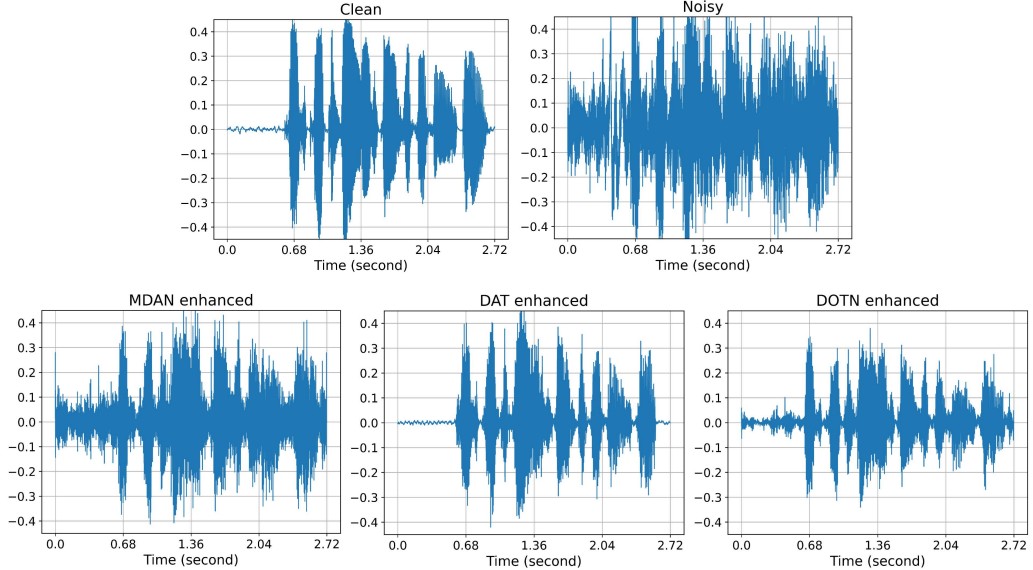

Figure 1: Waveforms for an utterance pronounced by a male speaker in Voice Bank (no.232) contaminated with Cafe background provided by DEMAND at SNR level 0 dB and its clean reference and enhanced versions.

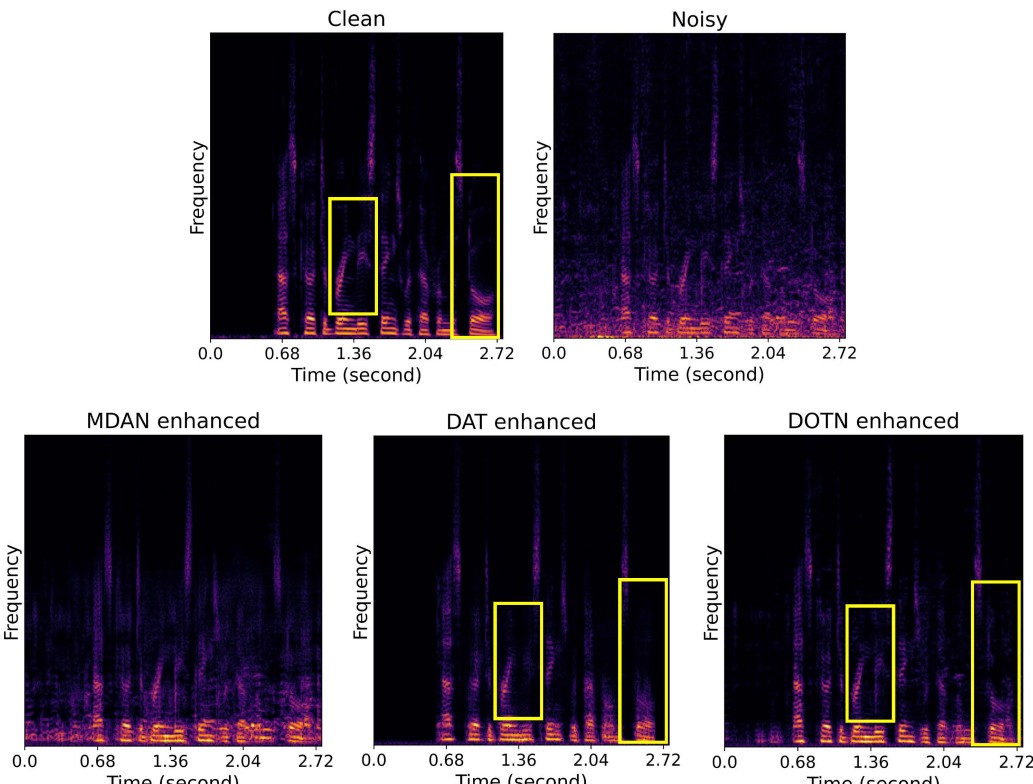

Figure 2: Spectrograms for an utterance pronounced by a male speaker in Voice Bank (no.232) contaminated with Cafe background provided by DEMAND at SNR level 0 dB and its clean reference and enhanced versions.

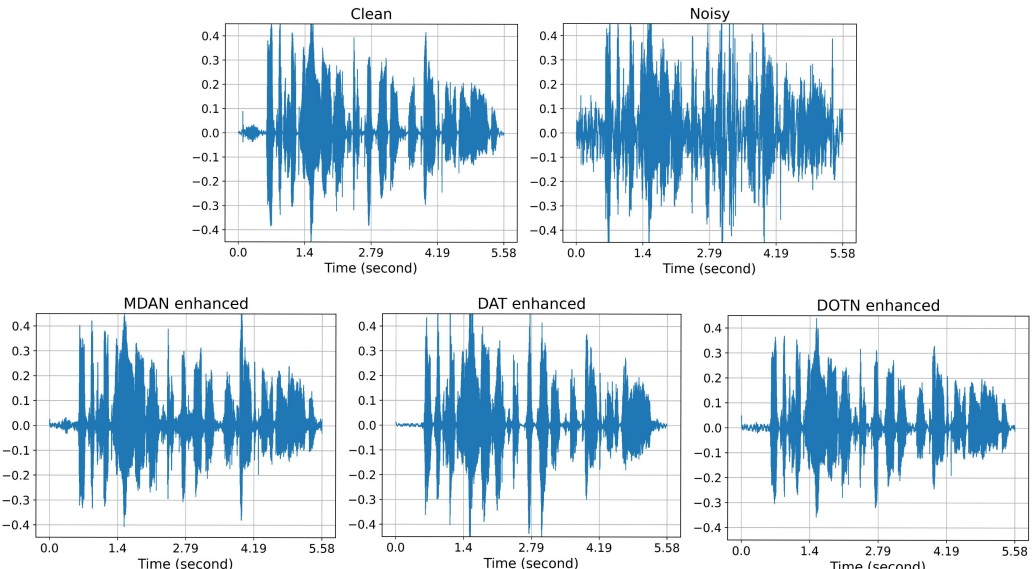

Figure 3: Waveforms for an utterance pronounced by a female speaker in Voice Bank (no.257) contaminated with Cafe background provided by DEMAND at SNR level 0 dB and its clean reference and enhanced versions.

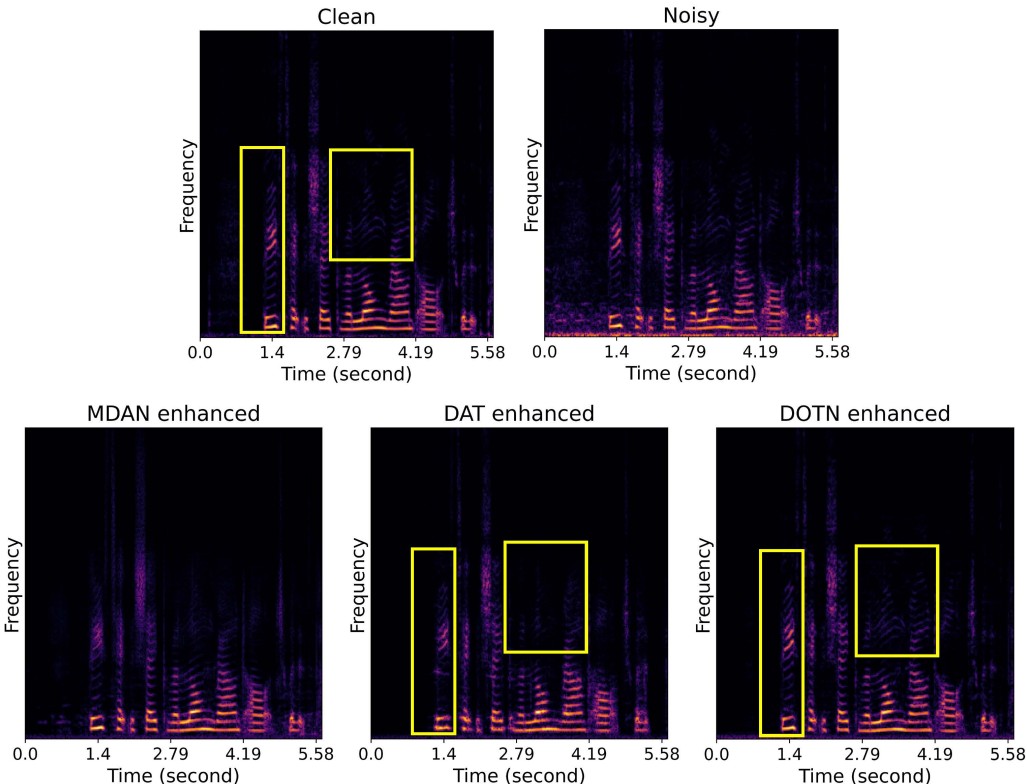

Figure 4: Spectrograms for an utterance pronounced by a female speaker in Voice Bank (no.257) contaminated with Cafe background provided by DEMAND at SNR level 0 dB and its clean reference and enhanced versions.

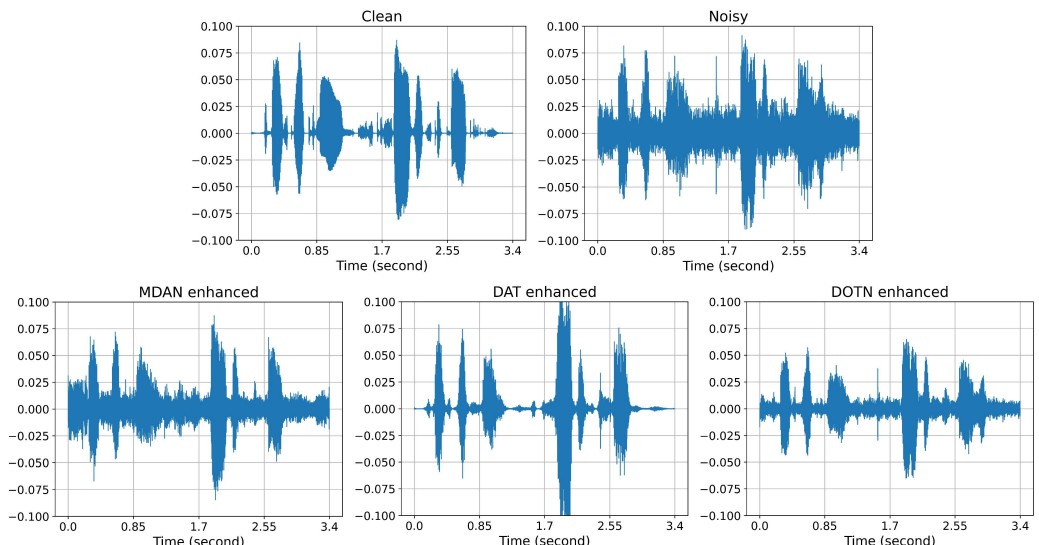

Figure 5: Waveforms for an utterance pronounced by a male speaker in TIMIT (labeled MTLS0) contaminated with Cafeteria background at SNR level 0 dB and its clean reference and enhanced versions.

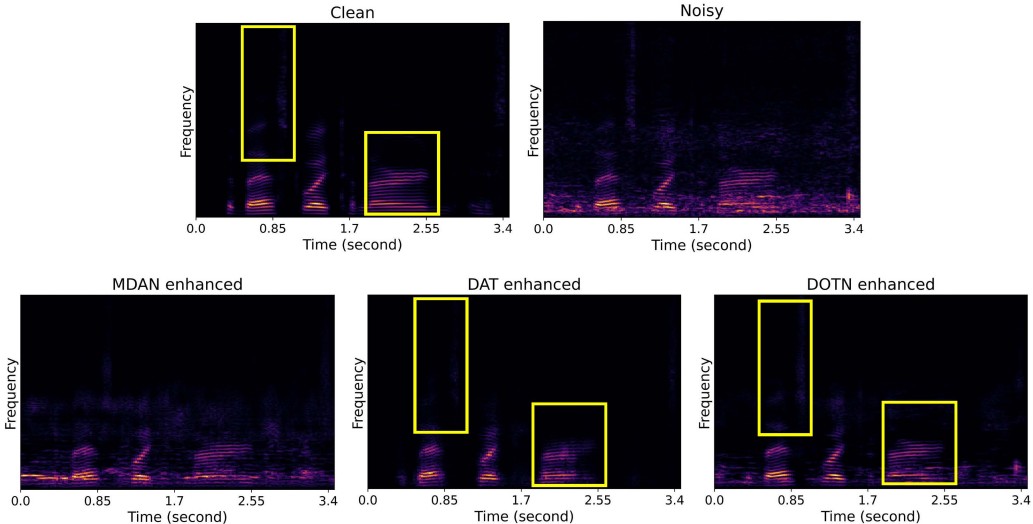

Figure 6: Spectrograms for an utterance pronounced by a male speaker in TIMIT (labeled MTLS0) contaminated with Cafeteria background at SNR level 0 dB and its clean reference and enhanced versions.

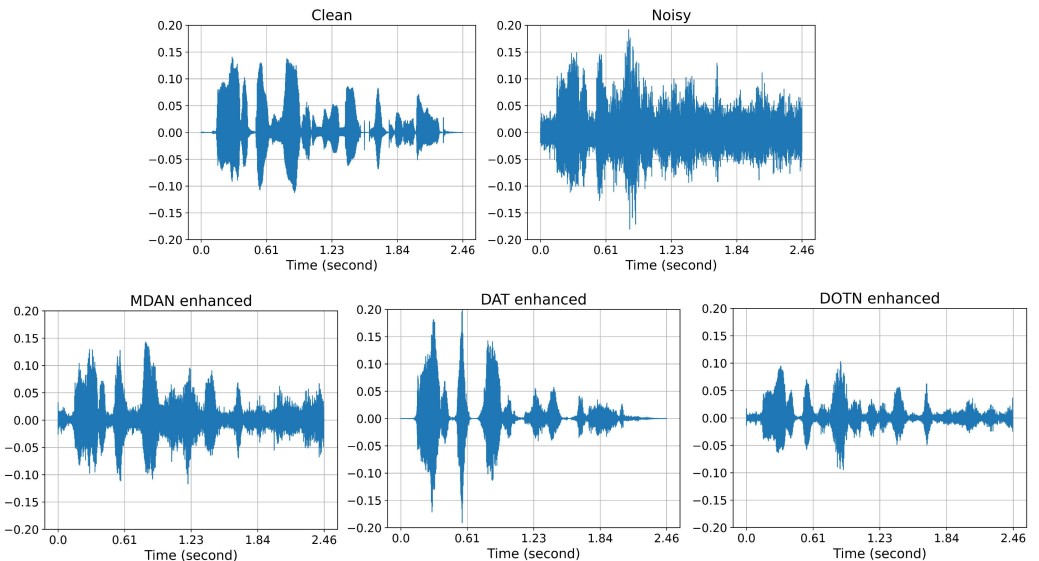

Figure 7: Waveforms for an utterance pronounced by a female speaker in TIMIT (labeled FDHC0) contaminated with Cafeteria background at SNR level 0 dB and its clean reference and enhanced versions.

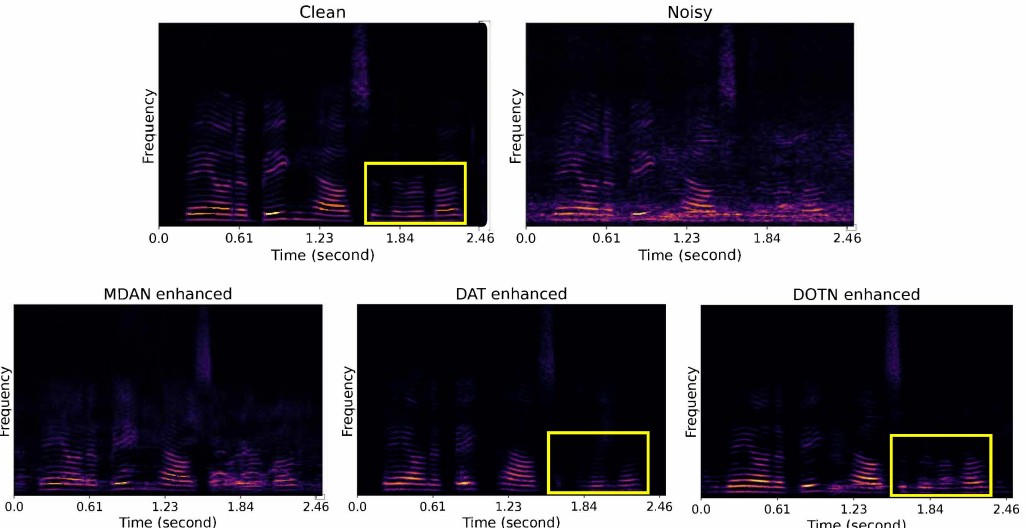

Figure 8: Spectrograms for an utterance pronounced by a female speaker in TIMIT (labeled FDHC0) contaminated with Cafeteria background at SNR level 0 dB and its clean reference and enhanced versions.

training renders a highly nonlinear constraint in the main OT alignment process to guarantee certain 'similarity' between the enhanced (fake) and clean (real) speech. One may question if it is appropriate to compare enhanced *target* data and clean *source* data. However, it is worth emphasizing that the task of discriminator is to roughly capture the character of natural clean speech, instead of making precise prediction in speech pattern. Due to the nature of this task, it does not post a logical issue when considering data in different domains.

## 2.2 Subjective evaluations

Subjective evaluations were conducted to collect individual opinions from their own perspectives, in contrast to formulated or objective metrics. The evaluation aimed to compare the proposed method with two weakly supervised method DAT and MDAN under human perceptions. 31 participants were gathered for blind test under random shuffles of audio recordings. All three enhancing methods appear in random orders; no knowledge of the audio source can be gained in advance. Participants were asked to rate each enhanced audio from 1 (bad) to 5 (excellent) for Mean Opinion Score (MOS) measurement. Only negative SNRs (dB) were used for demonstrating the significance of models, as well as reducing test time duration for participants.

For TIMIT, two denoised recordings were randomly chosen from 3 SNRs $(-3, -6, -9)$, 4 target noises ("helicopter", "crowd-party", "cafeteria", "babycry") and 3 models (DAT, MDAN, and DOTN), which amounts to 72 audio recordings for TIMIT. Similarly in VCTK, two denoised recordings were randomly selected among 3 SNRs $(-3, -6, -9)$, 3 target noises ("traffic", "CAFE", "public square") and 3 models (DAT, MDAN, DOTN), that amounts to 54 audio recordings. The MOS results of TIMIT and VCTK are listed Tables 1 and 2, respectively.

The results in Table 1, 2 showed that the human evaluations mostly favor DOTN over the other two methods. It is particularly dominant in the case of VCTK, Table 2, to confirm the perceptual performance of DOTN.

Table 1: MOS for TIMIT enhanced speech, ratings from 1 (bad) to 5 (excellent) for each audio recording given subjectively by each participant.

| noise type | Helicopter | | | Crowd-party | | |
|---|---|---|---|---|---|---|
| SNR/model | DAT | MDAN | DOTN | DAT | MDAN | DOTN |
| -9 dB | 2.08 | 2.03 | **2.16** | 1.84 | 1.47 | **2.13** |
| -6 dB | **2.76** | 2.24 | 2.66 | 1.89 | 1.90 | **2.68** |
| -3 dB | 3.10 | 2.32 | **3.11** | 2.68 | 2.24 | **3.10** |
| Avg | **2.65** | 2.20 | 2.64 | 2.14 | 1.87 | **2.64** |
| noise type | Cafeteria | | | Babycry | | |
| SNR/model | DAT | MDAN | DOTN | DAT | MDAN | DOTN |
| -9 dB | 1.84 | 1.47 | **2.61** | **2.71** | 1.58 | **2.71** |
| -6 dB | 2.24 | 1.61 | **2.74** | 3.26 | 2.37 | **3.47** |
| -3 dB | 2.56 | 2.19 | **3.35** | **3.40** | 2.60 | 3.37 |
| Avg | 2.21 | 1.76 | **2.90** | 3.12 | 2.18 | **3.18** |

Table 2: MOS for VCTK enhanced speech under the same setting as used in TIMIT.

| noise type | Traffic | | | Cafe | | | Public square | | |
|---|---|---|---|---|---|---|---|---|---|
| SNR/model | DAT | MDAN | DOTN | DAT | MDAN | DOTN | DAT | MDAN | DOTN |
| -9 dB | 2.10 | 2.05 | **3.68** | 2.15 | 1.76 | **3.89** | 3.05 | 2.44 | **3.85** |
| -6 dB | 2.56 | 2.81 | **4.11** | 2.35 | 2.19 | **3.61** | 2.74 | 2.52 | **4.00** |
| -3 dB | 3.19 | 2.76 | **4.23** | 3.40 | 2.47 | **4.34** | 3.76 | 3.29 | **4.29** |
| Avg | 2.62 | 2.54 | **4.01** | 2.63 | 2.14 | **3.95** | 3.18 | 2.75 | **4.05** |

## 2.3 Comparisons with the state-of-the-art supervised SE methods

Due to the lack of fully unsupervised domain adaptation methods for comparison with DOTN, an attempt to compare with the state-of-the-art (SOTA) "supervised" SE models may still be conducted to reveal some interests. A Transformer [6] model was used in the attempt of such comparison. A Transformer was trained from scratch on the datasets given in Sec. 4.2. Without domain adaptation, the Transformer as a supervised SE method was directly tested on the target domain. Table 3 showed the results with all SNRs summed over in three target domains.

It was observed that the DOTN had slightly better performance over the Transformer in most metrics. This result may be expected as the Transformer as a supervised SE method did not contain an adaptation mechanism to well adjusted to the target domain, in which case the background noise types were Cafe, Traffic, and Public Square.

Table 3: Comparisons to a state-of-the-art method in SE.

| noise/metric | Transformer PESQ/STOI | DOTN PESQ/STOI |
|---|---|---|
| Cafe | 2.225/0.791 | **2.244/0.803** |
| Traffic | 2.496/0.840 | **2.525/0.850** |
| Public Square | **2.610**/0.858 | 2.577/**0.865** |

On the first glimpse, this does not seem a meaningful comparison, as the Transformer is supervised, while the proposed DOTN is designed for unsupervised adaptations. However, the comparison is based on the same training and testing sets, and the better performance of DOTN verifies the advantage of adaptation process. This advantage of adaptation should be more evident when we restrict the training set to be smaller. Another SOTA supervised SE method MetricGAN+ [7] can be used to conduct an additional comparison as the Transformer here. The full table and discussion can be found at `https://drive.google.com/drive/folders/1cO3GCeFnQVXpatKXoyIO-b_PvfZwq6hO?usp=sharing`.

## 2.4 An example on real-world applications

We also applied the adaptation models (DAT, MDAN, DOTN) to a real-world noisy speech data CHiME-3 for comparison; the enhanced audios can be found here: `https://drive.google.com/drive/folders/1C4BiajlZfMjBXbemTtiA8PUIZpVhmQ4O?usp=sharing`.

The audio demos consist of speech from two random male and female speakers selected from three noisy environments (Cafe, Street, Pedestrian). The DAT and MDAN models applied here on CHiME-3 were previously trained by adapting from the source noise domain: "Bus", "Car", "Metro" to target noise: "Traffic" under VCTK-DEMAND, as in the Sec. 4.2 of the paper.

As there is no clean speech for reference, DNSMOS [8] scores are computed alternatively for quality measures. The DNSMOS scores, provided here, averaging over audio samples are:

$$\text{noisy} : 2.73, \quad \text{DAT} : 3.12, \quad \text{MDAN} : 2.99, \quad \textbf{DOTN} : \textbf{3.27}$$

A possible reason for the degraded performance in DAT, MDAN compared to DOTN may be that the target domains appeared in CHiME-3: Cafe, Street, Pedestrian, did not appear in their pretrained target categories in the first place. *i.e.,* their domain classifiers had not seen such type of noise, and hence the domain mismatch may remain large.

On the other hand, the proposed DOTN aligns one domain to another by OT so that the data origin is not required. Many downstream tasks can therefore be achieved, especially when a new target domain without additional information on labels is confronted.