# OpenReview forum: "Unsupervised Noise Adaptive Speech Enhancement by Discriminator-Constrained Optimal Transport"
_NeurIPS.cc/2021/Conference — NeurIPS 2021 Poster_

### Official Review · Reviewer_8JUP · 2021-07-14

**Rating:** 6
**Confidence:** 4

**Summary:**

This paper proposes using a discriminator-constrained optimal transport method for unsupervised speech enhancement (SE). The method is specifically designed for a regression task. It involves 2 parts. The first parts estimates a generator function as an OT problem to align the source and target labels. The source-target alignments ($\gamma$) and the generator function parameters (${\theta}_f$) are estimated in an interleaving fashion. The second part incorporates a discriminative training component on the label distributions using WGAN. Two discriminative loss functions were used, one for the generator (${\cal L}_f$) and another for the discriminator (${\cal L}_h$). The discriminator losses are applied periodically at a pre-defined interval ($n_f$ and $n_h$).

The proposed method is referred to as the discriminative-constrained optimal transport network (DOTN). It does not require any additional knowledge about the noise types during training. This paper compares DOTN with two other adversarial domain adaptation methods, namely DAT and MDAN, using two datasets (VoiceBank and TIMIT). Different types of noise are artificially added to these datasets using the DEMAND dataset at different SNR levels to create different noise domains. The quality of the speech enhancement methods are evaluated using the PESQ and STOI metrics.

The overall results show that the proposed DOTN outperforms DAT and MDAN on both datasets. However, the model sizes and computational complexities of the models are not reported, making it difficult to compare different methods. Besides, there is lack of detailed discussions/analyses about the design of the training algorithm (see main review).





**Limitations And Societal Impact:**

The results reported in the paper are based on artificially created noisy speech data. It is unclear how well the same method will perform on realistic noisy data. Furthermore, the lack of details about the models being evaluated makes it difficult to make a fair comparison.


**Main Review:**

Originality:

The use of optimal transport (OT) for adversarial domain adaptation has been investigated in prior work for classification tasks. This work focuses on applying it to speech enhancement, which is a regression problem. However, the paper did not discuss the specific challenges in applying OT domain adaptation to a regression problem.

Quality:

The experiments are limited to comparing the overall SE performance of DOTN, DAT and MDAN on two datasets. Some important details are missing and more ablation studies could have been done. For example:

- How important is it to include the discriminator training stage?
- How were the hyper-parameters ($\alpha$, $\beta$, $n_f$ and $n_h$) tuned? How sensitive were they on speech enhancement performance?

Clarity:

The paper is generally clear enough to understand and follow through. Given that the paper focuses on comparing DOTN with DAT and MDAN, it can be useful to include a section that compare and contrast these methods. Some minor comments:
- In Algorithm 1, $t$ is used in lines 4 & 7. So, it should be included in the **Require:** section.
- In Algorithm 1, in lines 5 & 8, both $\theta_f$ and $\theta_h$ should be passed to $Adam(...)$ since both $f(.)$ and $h(.)$ are used to compute ${\cal L}_f$ and ${\cal L}_h$.
- In Eq. 1, is it necessary to include the $\alpha || x_i^s - x_j^t||^2$ term (since it is independent of $\gamma$ and $f$)?

Significance:

The problems being investigated in this paper is specific to speech enhancement (or regression problem in general). The experimental setup is limited to adapting to different noise domains and the noisy data are obtain by artificially adding noise to clean datasets. The study would have been more impactful if realistic noisy data were used.

---

After the rebuttal: I have read the rebuttal by the authors. Given the new results provided on CHiME3 dataset, I have increase my score by 1.

**Time Spent Reviewing:**

2

---

> ### Author Response · Authors · 2021-08-10
> **Reply to Reviewer 8JUP**
>
> We thank the reviewer for the detailed feedback and valuable suggestions. Our responses following the corresponding comments are organized point-by-point as follows:
>
> > **[Originality] The use of optimal transport (OT) for adversarial domain adaptation has been investigated in prior work for classification tasks. This work focuses on applying it to speech enhancement, which is a regression problem. However, the paper did not discuss the specific challenges in applying OT domain adaptation to a regression problem.**
>
> - The challenges mainly came from two parts:
>    1. Different from classification problems where outcomes are only categorical, regressions demand prediction accuracy in a continuous fashion, in that a regression problem is an extreme case where the classification has infinite categories. Therefore, regression problems are in general more difficult than classification problems. Due to this fundamental difference, methods that succeed in classifications are usually not directly applicable to regressions.
>    2. Another difficulty came from that even if a regression architecture manages to be setup properly and trained successfully by observing decreasing losses, the regression results may not always be as desired. As was observed in speech enhancement tasks, the outcome speech may be highly distorted or twisted. As there are more subtle/fine structures for human to recognize it as a natural voice. Thus, the challenge includes not only elevating the accuracy level continuously, but also capturing the essential characteristics of the learning objects.
>
>    This comment is much appreciated and a more detailed discussion shall be included for the challenges encountered in applying OT domain adaptation to regression problems.
>
>
> >  **[Quality] How important is it to include the discriminator training stage?**
>
> - This is a good question where the discriminator training indeed plays a critical part for the success of our OT domain adaptation (DA) model. In fact, our first attempt of applying plain OT DA to the speech enhancement (SE) was not satisfactory as we faced the challenges mentioned above. This predecessor essentially followed the tradition to rely on Mean-Square loss for domain alignment. Later we came to realize that it was insufficient for the complex characteristics of human speech. A subtle and refined criterion was then sought to help forming natural voices; this is where a discriminator came in for assistance. Consequently, a discriminator was introduced to govern the similarity of target output and clean speech in a highly nonlinear manner (compared to MSE) so that the high quality of speech output can be guaranteed.
>
>
> > **[Quality] How were the hyper-parameters ($\alpha$, $\beta$, $n_h$, $n_f$) tuned? How sensitive were they on speech enhancement performance?**
>
> - The hyper-parameters were empirically tuned on the validation set. Our experience showed that SE performance was not sensitive on the choice of $\alpha$, $\beta$ as long as the input ($\alpha || |x^s - x^t||$) and output terms ($\beta || y - f(x)||$) maintain comparable magnitudes. On the other hand, $n_f$ and $n_h$ affect performances noticeably, and the optimal parameters vary across datasets and noise types, as reported in some articles regarding GANs.
>
> > **[Clarity] Given that the paper focuses on comparing DOTN with DAT and MDAN, it can be useful to include a section that compare and contrast these methods.**
>
> - We thank the reviewer for this helpful suggestion. In addition to the technical details of the DAT and MDAN architectures submitted in the supplementary materials earlier, we agree that an additional section for method comparison will improve the manuscript. It will be appended to the manuscript once the revision is allowed.
>
> > **[Clarity]**
> >  - **In Algorithm 1, $t$ is used in lines 4 & 7. So, it should be included in the Require: section.**
> >  - **In Algorithm 1, in lines 5 & 8, both $\theta_f$ and $\theta_h$ should be passed to $Adam(\cdots)$ since both $f(\cdot)$ and $h(\cdot)$ are used to compute $\mathcal{L}_f$ and $\mathcal{L}_h$.**
> - Thank you very much for pointing out the issues in Algorithm 1. They will be corrected when the manuscript is allowed to be revised.
>
>
> >  **In Eq. 1, is it necessary to include the term $\alpha || x_i^s - x_j^t ||^2$ (since it is independent of $\gamma$ and $f$)?**
> - We understand the confusion, but yes, the term is necessary. This term in the cost $C_{ij}$ will affect the consequent OT plan $\gamma = (\gamma_{ij})$ solved by minimizing $\sum_{i,j} \gamma_{ij} (\alpha ||x_i^s - x_j^t ||^2 + \beta || y_i^s - f(x_j^t) ||^2)$, so that changing $C_{ij}$ may result in wrong transportation plan $\gamma$. The purpose of $\alpha ||x_i^s - x_j^t ||^2$ in the cost matrix is to search for the most similar (labeled) source sample ($x_i^s$) given an unlabeled target sample ($x_i^s$).
>
>
> >  **[Significance] The experimental setup is limited to adapting to different noise domains and the noisy data are obtain by artificially adding noise to clean datasets. The study would have been more impactful if realistic noisy data were used.**
>
> - As this work is the first proposal of an unsupervised OT domain adaptation method for speech enhancement, we aim to introduce the methodology and support it with comprehensive standard testing, where the noisy datasets were generated using standard speech corpuses and various noise types at several well-controlled SNR levels. We agree that further testing with realistic noisy data will be valuable, and it will be one of future works for investigation.
>
> > **[Limitations And Societal Impact] The results reported in the paper are based on artificially created noisy speech data. It is unclear how well the same method will perform on realistic noisy data. Furthermore, the lack of details about the models being evaluated makes it difficult to make a fair comparison.**
>
> - We agree that further testing with realistic noisy data will be valuable. However, when testing performance using real-world noisy utterances, the corresponding clean utterances are generally inaccessible. It is noted that most standardized SE evaluation metrics require clean speech data as reference. Evaluation metrics without clean speech data as reference often show biased results, which may not be trustful. Since this work serves as a pioneering work of purely unsupervised adaptation for SE, we intended to compare different SE adaptation methods confidently so that readers can also easily quantify and reproduce our results. Therefore, in the current study, we only conducted experiments on two standardized datasets, TIMIT and VoiceBank-DEMAND, where clean speech data are available. Accordingly, standardized and popular evaluation metrics, such as PESQ and STOI, can be computed.
>
>   This suggestion is very helpful; we are currently design another experiment on realistic noisy data and shall update this question as soon as possible.

---

> > ### Author Response · Authors · 2021-08-19
> > **Experiments on realistic noisy data CHiME3**
> >
> > > **[Limitations And Societal Impact] The results reported in the paper are based on artificially created noisy speech data. It is unclear how well the same method will perform on realistic noisy data. Furthermore, the lack of details about the models being evaluated makes it difficult to make a fair comparison.**
> > - **[Addendum]** On reviewer's suggestions, we have applied the adaptation models (DAT, MDAN, DOTN) to a real-world noisy speech data CHiME-3, which was originated from a challenge http://spandh.dcs.shef.ac.uk/chime_challenge/chime2015/. The enhanced audios are now appended in the supplemental material: https://drive.google.com/drive/folders/1xJJtVyVAkEdL6Absm63bB3sIk7VnECal. The audio demos consist of speech from two random male and female speakers selected among 3 noisy environments (Cafe, Street, Pedestrian) out of 4 (Cafe, Street, Pedestrian, Bus). The three models applied were previously trained from the VCTK-DEMAND Transportation category: "Bus", "Car", "Metro" to adapt "Traffic" of the Street category, as described in Sec. 4.1. As there is no clean speech for reference, DNSMOS scores [1] are computed as an alternative for quality measure: https://drive.google.com/file/d/1meNpEhkn-oLyxVYNuSrw2uNQtv7oz5iF/view. The DNSMOS scores averaging over audio samples are as the following:
> >
> >
> >     noisy: 2.73, DAT: 3.12, MDAN: 2.99, DOTN: 3.27
> >
> >
> > Ref [1]: DNSMOS metric (https://arxiv.org/pdf/2010.15258.pdf)

---

### Official Review · Reviewer_Km9T · 2021-07-16

**Rating:** 8
**Confidence:** 4

**Summary:**

This paper presented a novel method to achieve unsupervised domain adaptation for speech enhancement. There is little work in unsupervised domain adaptation for regression tasks and the paper combined techniques from optimal transport theory and generative adversarial networks, accomplishing better results on two datasets, VoiceBank-DEMAND and TIMIT, while requiring less supervision than prior work.

**Limitations And Societal Impact:**

The authors adequately addressed limitations by discussing performance drops when too many sources of noises are present simultaneously. The paper do not have any immediate negative impacts. The only concern is that by improving the performance of various speech tasks, there can be more privacy violation or surveillance.

**Main Review:**

Originality: Although there exists many works on unsupervised domain adaptation, the authors proposed a novel method to tackle the speech enhancement task, which resembles regression rather than classification, an area that had seen much less success previously. Compared to prior work on unsupervised domain adaptation for speech enhancement, the proposed discriminator-constrained optimal transport network (DOTN) does not require any additional labels. The paper uses a novel combination of known techniques to solve an important problem. The related work and contributions were stated very clearly.

Quality: The authors demonstrated the strength of their novel approach well using two standard datasets (VoiceBank-DEMAND and TIMIT) and two metrics (PESQ and STOI), while comparing against two other prior work on unsupervised domain adaptation for speech enhancement. How the datasets were cleaned and prepared was described in detail and results and comparisons were analyzed in depth. The authors also tested and stated the limitations well by performing the multiple noise source experiment, testing the effects of catastrophic forgetting.

Clarity: The submission was very well written and clearly organized. One minor suggestion is to flip the row orders of noise type and methods in the tables to make comparisons clearer.

Significance: The results are quite significant as speech enhancement could benefit many downstream tasks and the paper was able to achieve better performance than prior works on unsupervised domain adaptation while not needing additional labels. Having a truly unsupervised setting is quite important for real world use since knowing the type of noise is an unrealistic assumption in many cases.

**Time Spent Reviewing:**

1.5

---

> ### Author Response · Authors · 2021-08-10
> **Reply to Reviewer Km9T**
>
> **To the Reviewer's comments:**
>
> > **Clarity: The submission was very well written and clearly organized. One minor suggestion is to flip the row orders of noise type and methods in the tables to make comparisons clearer.**
>
> - Thank you very much for your positive feedback and helpful comments. Indeed, it is our humble hope that our proposal first to achieve unsupervised domain adaptation can benefit related ML developments. The suggestion on flipping the row orders in our tables will be modified once the manuscript is allowed to be revised.

---

### Official Review · Reviewer_uPz2 · 2021-07-16

**Rating:** 5
**Confidence:** 4

**Summary:**

The authors propose an approach for speech enhancement based on optimizing the empirical OT loss between the joint distribution of noisy/clean training data and noisy/cleaned target data (c.f. (1), line 139). The cleaned speech outputs are additionally optimized under a standard WGAN formulation (c.f. line 129). Their approach consistently outperforms DAT[19] and MDAN[44] on VoiceBank-DEMAND and Helicopter-TIMIT mixtures, and performs roughly on-par or slightly worse on TIMIT-(Crowd-party, Cafeteria, Baby-cry) mixtures which contain speech, according to the standard PESQ and STOI metrics.

**Ethical Concerns:**

No.

**Limitations And Societal Impact:**

Yes.

**Main Review:**

Strengths:

- OT is a very natural formulation to domain adaptation, and has been adopted by others.
- Gains on non-speech noise over baselines.
- Code made available, so should be reproducible.

Limitations:

- Lower novelty, standard training metrics for domain adaptation.
- The baselines are not SOTA speech enhancement techniques, and the results are on non-standard datasets, reducing the significance of the results.
- Standard metrics are reported but no audio demonstrations or human evaluations (MOS) are provided, these are important ways to assess true performance.
- The approach does not improve mixtures containing other speech, which is another signficant limitation.

Current Assessment:

- Falls short wrt ML novelty for Neurips and significance to the ML and speech enhancement communities.

----

Post-rebuttal (updated)

Thank you to the reviewers for their detailed responses.
- The audio demonstrations reveal that all of the methods compared are performing quite poorly, far below SOTA.
- (update 1) The follow-up MetricGAN+ (SOTA) results provided show that the approach achieves slightly better STOI scores, but significantly worse PESQ scores. However, the fact that the MetricGAN+ is optimized on PESQ makes this comparison more difficult to interpret.
- (update 2) Follow-up MetricGAN+ and Transformer audio demos are subjectively significantly better than the proposed method. MOS or A/B preference listening tests vs. MetricGAN+ and/or the SE Transformer should be included in the results. The current audio demos suggest that the method would be outperformed by MetricGAN+ and the SE Transformer in terms of MOS or A/B assessment, but only 2 audio examples were provided.
- The manuscript needs substantial revision. The paper in general, and particularly the experimental section is not detailed enough to establish the contributions the paper makes, and does not describe the approach, particularly in mapping abstract losses and inputs/outputs to a concrete system and associated details.
- As a type of kernel-based method for adaptation/reconstruction, the training sets are likely too small, and the details regarding reconstruction (chunking etc.) likely not fully optimized, perhaps explaining the results.
- (update 3) Based on the additional results provided, I have increased my score, but remain concerned that the manuscript and results are not complete enough to justify an acceptance recommendation.

**Time Spent Reviewing:**

3

---

> ### Author Response · Authors · 2021-08-10
> **Reply to Reviewer uPz2**
>
> We thank the reviewer for the constructive feedback and valuable suggestions. Our point-by-point comments are as follows:
>
> > **Lower novelty, standard training metrics for domain adaptation.**
> - Our novelty lies in:
>      1. First to achieve purely unsupervised domain adaptation for complex regression tasks (especially Speech Enhancement) to our best knowledge.
>      2. Able to relinquish the condition on additional target information. Moreover, our method outperforms current algorithms that require extra target label information, and thus is more helpful for real-world applications.
>
> > **The baselines are not SOTA speech enhancement techniques, and the results are on non-standard datasets, reducing the significance of the results.**
> -  It is our understanding that the fully-unsupervised SOTA for SE domain adaptation has not been developed yet, no proper work may be directly compared. Therefore, two most related methods of weak supervision were used for equal-footing comparisons.
> On the other hand, VCTK and TIMIT are frequently used datasets for SE evaluations. (see e.g., https://arxiv.org/pdf/1703.09452.pdf [VCTK-DEMAND], https://ieeexplore.ieee.org/document/6932438 [TIMIT]) only the noise mixture procedure is pertaining to our own customization. For clarity our noise mixture code will later be released to Github for comparison as well.
>
>
> > **Standard metrics are reported but no audio demonstrations or human evaluations (MOS) are provided, these are important ways to assess true performance.**
>
> - 1. Some audio demonstrations can be found in the original supplementary material submitted: https://drive.google.com/drive/folders/1k66yW6v6AoKYJ7PVHlxadnKvXVauIVcV?usp=sharing
>   2. To include human evaluations is a great advice as another indicator, the results are as follows: https://drive.google.com/file/d/1xQfR0pR66BhVYYyyyLdc9xVqzdBUfb_m/view?usp=sharing. We managed to conduct human evaluations (MOS) as suggested: 31 persons were gathered for blind test by random shuffles of audio recordings. Only negative dBs were conducted for showing the significance of models as well as reducing testing time duration of participants. For TIMIT and VCTK, two denoised recordings were randomly chosen for each SNR (-3, -6, -9), each noise type (e.g. “helicopter”, “cafeteria”, … ), and each model (DAT, MDAN, DOTN), which amounts to 72 audio recordings for TIMIT and 54 audio recordings for VCTK. As the results show, the human evaluations mostly favor DOTN over the other two models. It is particularly dominant in the case of VCTK to confirm the performance of our method. More details please refer to the table provided in the link.
>
>
> > **The approach does not improve mixtures containing other speech, which is another significant limitation.**
> -  Slight degraded results under speech mixture are logically expected due to the purely unsupervised manner we seek. This is because no additional information is provided to distinguish the main speech from another when the noise comes from other speeches. As pointed out in *line 224*, both DAT and MDAN can use additional domain information to decouple mixtures and suppress another speech as noise, but the purely unsupervised DOTN does know a speech mixture situation in advance. In this view, a certain degree of degradation on mixing speech appears to be an unavoidable trade-off at this point for the purely unsupervised domain adaptation. Even so, please note that our method still achieves comparable or even superior speech enhancement performance in numerous testings under a purely unsupervised setting.

---

> ### Author Response · Authors · 2021-08-31
> **Reply to new comments after post-rebuttal**
>
> Thank you very much for your comments. We would like to provide the following additional information:
> > **The audio demonstrations reveal that all of the methods compared are performing quite poorly, far below SOTA.**
>
> - Please note that our test set was designed under a relatively challenging scenario (low SNR and non-stationary noise types), so that it became much more challenging than the standard VoiceBank-DEMAND testing set. The comparison with the state-of-the-art **MetricGAN+ [1]** (derived from *Speechbrain: https://huggingface.co/speechbrain/metricgan-plus-voicebank*) and **Transformer [2]** speech enhancement approaches on our test set was made, as the enhanced audio samples can be found here: https://drive.google.com/drive/folders/1PC10hN9uCwVylP-LCDykdbr8MBi60VTs
> The audio samples demonstrated that the enhanced sentences by the proposed DOTN model may be not as clean as those processed by MetricGAN+ and Transformer but preserve more speech components and possess more natural tone.
>
> - Implementing the state-of-the-art **(SOTA) MetricGAN+** approach on our testing set derived the overall average metrics over different SNRs **PESQ=2.829, STOI=0.791** for the Cafe environment, while **DOTN** yielded **PESQ=2.244, STOI=0.803 (by Table 1 and 2)**. The results then showed that the proposed DOTN (with unsupervised model adaptation) can yield comparable and sometimes even better performance as compared to the SOTA MetricGAN+ in terms of intelligibility scores.
>
> - We are grateful for the reviewer providing this constructive comment. We will include the results of MetricGAN+ on our dataset to clarify that the dataset is relatively challenging.
>
>
>     [1] Fu, S. W., Yu, C., Hsieh, T. A., Plantinga, P., Ravanelli, M., Lu, X., & Tsao, Y. (2021). MetricGAN+: An Improved Version of MetricGAN for Speech Enhancement. arXiv preprint arXiv:2104.03538.
>
>     [2] Koizumi, Y., Yatabe, K., Delcroix, M., Masuyama, Y., & Takeuchi, D. (2020, May). Speech enhancement using self-adaptation and multi-head self-attention. In ICASSP 2020-2020 IEEE International Conference on Acoustics, Speech and Signal Processing (ICASSP) (pp. 181-185). IEEE.
>
> > **The manuscript needs substantial revision. The paper in general, and particularly the experimental section is not detailed enough to establish the contributions the paper makes, and does not describe the approach, particularly in mapping abstract losses and inputs/outputs to a concrete system and associated details.**
>
> - Thank you for pointing this out. Indeed, more experimental details will certainly be amended to the manuscript when further revision is allowed.
>
> > **As a type of kernel-based method for adaptation/reconstruction, the training sets are likely too small, and the details regarding reconstruction (chunking etc.) likely not fully optimized, perhaps explaining the poor results. I encourage the authors to revise and resubmit their work to a future conference.**
>
> - Please note that the main focus of this study is to demonstrate the effectiveness of the proposed unsupervised SE model adaptation approach. We adopted a well-known BLSTM model to build the SE model. The model architecture and design (chunk size) have been well-tuned on the TIMIT and standardized Voice Bank-Demand datasets. The readers can easily reproduce our system. We also show that our system can yield comparable or even better performance as compared to SOTA MetricGAN+. Importantly, the proposed method can provide promising results even with limited amount of training data.
>
>  **[Additional Evidence]** On reviewer 8JUP's suggestions, we have applied the adaptation models (DAT, MDAN, DOTN) to a real-world noisy speech data CHiME-3, which was originated from a challenge http://spandh.dcs.shef.ac.uk/chime_challenge/chime2015/. The enhanced audios are now appended in the supplemental material: https://drive.google.com/drive/folders/1xJJtVyVAkEdL6Absm63bB3sIk7VnECal. The audio demos consist of speech from two random male and female speakers selected among 3 noisy environments (Cafe, Street, Pedestrian) out of 4 (Cafe, Street, Pedestrian, Bus). The three models applied were previously trained from the VCTK-DEMAND Transportation category: "Bus", "Car", "Metro" to adapt "Traffic" of the Street category, as described in Sec. 4.1. As there is no clean speech for reference, DNSMOS scores [1] are computed as an alternative for quality measure: https://drive.google.com/file/d/1meNpEhkn-oLyxVYNuSrw2uNQtv7oz5iF/view. The DNSMOS scores averaging over audio samples are as the following:
>
>
>     noisy: 2.73, DAT: 3.12, MDAN: 2.99, DOTN: 3.27
>
>
> Ref [1]: DNSMOS metric (https://arxiv.org/pdf/2010.15258.pdf)

---

> > ### Comment · Reviewer_uPz2 · 2021-09-02
> > **PESQ, STOI results for the MetricGAN+ [1] and Transformer [2] approaches**
> >
> > Authors, thank you for the additional information. Can you provide more complete PESQ and STOI results for the MetricGAN+ [1] and Transformer [2] approaches? Human A/B preference or MOS scores would be even better of course, but I realize that this will take time... Also, can you comment on the data used to train these systems, and anything else that should be taken into account when comparing the results (i.e., are they fair comparisons?). Many thanks!

---

> > > ### Author Response · Authors · 2021-09-03
> > > **Reply to PESQ, STOI results for the MetricGAN+ [1] and Transformer [2] approaches**
> > >
> > > Thank you for the inquiry, we are pleased to provide more information on comparisons of MetricGAN+ and Transformers. Currently we are working on computing all other scores for complete results. Latest information shall be updated as soon as possible. Also, the VCTK-DEMAND dataset is used to train MetricGAN+ and Transformer, exactly the same as the ones to compute Table 1 & 2 in the manuscript with details described in Sec. 4.1, so that this would provide a fair ground to compare the results.

---

> > > > ### Author Response · Authors · 2021-09-04
> > > > **Quick updates to comparison with SOTA MetricGAN+**
> > > >
> > > > Due to the limited time constraint, we are still working on training MetricGAN+ and Transformer as designed earlier. A quick comparison with **pretrained** (SOTA) MetricGAN+ is performed instead for the purpose of timeliness:
> > > >
> > > > | PESQ/STOI   | MetricGAN+    | DOTN  |
> > > > | :------------- :|:-------------:| :-----:|
> > > > | CAFE         |  **2.829**/0.791  | 2.244/**0.803** |
> > > > | TRAFFIC   |  **3.111**/0.846  | 2.525/**0.850** |
> > > > | SQUARE   |  **3.082**/0.861  | 2.577/**0.865** |
> > > >
> > > > This table can be glimpsed that the MetricGAN+ has higher PESQs, while DOTN has higher STOIs over MetricGAN+. However, we point out that this comparison may not be fully indicative as the pretrained MetricGAN+ model was exposed to the noise types of the target domain (CAFE, TRAFFIC, & SQUARE). Of course, an equal-footing comparison is still under way.
> > > >
> > > > The numerical values verify that the high PESQs in MetricGAN+ are due to the design of direct PESQs optimization into objective functions.
> > > >
> > > > Since MetricGAN+ and DOTN share similar designs utilizing the GAN architecture, the cooperation of direct PESQ optimization into objective functions of DOTN may be a future direction to be investigated. The discussion of "the cooperation of PESQ optimization into DOTN will be included in the conclusion.

---

> > > ### Author Response · Authors · 2021-09-14
> > > **Reply to post-rebuttal latest updates**
> > >
> > > > **(update 1) The follow-up MetricGAN+ (SOTA) results provided show that the approach achieves slightly better STOI scores, but significantly worse PESQ scores. However, the fact that the MetricGAN+ is optimized on PESQ makes this comparison more difficult to interpret.**
> > >
> > > - Indeed, direct optimization of MetricGAN+ aiming high PESQs would be hard to interpret. On the other hand, the comparison of DOTN with a train-from-scratch Transformer on the same datasets given in Table 1 & 2 of Sec. 4 is shown below:
> > >
> > > | PESQ/STOI   | Transformer    | DOTN  |
> > > | :------------- :|:-------------:| :-----:|
> > > | CAFE         |  2.225/0.791  | **2.244**/**0.803** |
> > > | TRAFFIC   |  2.496/0.840  | **2.525**/**0.850** |
> > > | SQUARE   |  **2.610**/0.858  | 2.577/**0.865** |
> > >
> > > - The observation indicates that the DOTN performs better in general than the Transformer under same training settings (same epochs and same learning rates), and thus addressed the concern of comparable capabilities with current SOTA methods.
> > >
> > >
> > > > **(update 2) Follow-up MetricGAN+ and Transformer audio demos are subjectively significantly better than the proposed method. MOS or A/B preference listening tests vs. MetricGAN+ and/or the SE Transformer should be included in the results. The current audio demos suggest that the method would be outperformed by MetricGAN+ and the SE Transformer in terms of MOS or A/B assessment, but only 2 audio examples were provided.**
> > >
> > > - This comment is much appreciated. We humbly believe that in addition to speech quality, the speech intelligibility also serves as an important indicator to the speech enhancement performance. From our experimental results, DOTN yielded higher STOI scores as compared to MetricGAN+ and Transformer.
> > >
> > >   On the other hand, the two audio examples provided in the Google Drive link were intended to promptly answer one of another reviewer’s questions. More audio samples are intended to be released by constructing a complete Github website for showcase.
> > >
> > >
> > > > **(update 3) Based on the additional results provided, I have increased my score, but remain concerned that the manuscript and results are not complete enough to justify an acceptance recommendation.**
> > >
> > > - Thank you for increasing the score and seeing the value of our proposed work. We understand that the Reviewer had concerns over experimental details not fully addressed in the manuscript and also the novelty of applying OT.
> > >
> > >   However, since the DOTN code was available along with other methods: DAT, MDAN, technical comparisons can be observed through running codes. Also, the implementation details considered important by the reviewers’ suggestions will be organized into the manuscript or the supplementary material later whenever possible.
> > >
> > > - Lastly, we like to emphasize that DOTN is the first work to attack unsupervised domain adaptation problems for complex regression tasks in our understanding. Previous studies focused on classification tasks. As is known, regression problems are in general more demanding than classifications in terms of predictions. Therefore, there exist certain difficulties for the unsupervised regression domain adaptation to work properly. It is our humble hope to report and share our findings and the implementation details to contribute to the machine learning society.

---

### Official Review · Reviewer_Ryjo · 2021-07-20

**Rating:** 6
**Confidence:** 3

**Summary:**

The paper presents an unsupervised noise adaptive speech enhancement method by combining optimal transport network and Wasserstein generative adversarial network. Experimental results show that the method is effective.

**Limitations And Societal Impact:**

Yes

**Main Review:**

The main novelty of this work is to propose a discriminator-constrained optimal transport network for unsupervised domain adaptation for speech enhancement. It is unsupervised and outperforms two weakly-supervised methods. The written of the paper is generally okay but the clarity could be improved as exemplified below.

Optimal transport (OT) is the theory fundamental to this work. However, it is neither well explained (thinking OT is less known for the speech enhancement community) nor well motivated (namely why it is a good solution for this problem).

The abbreviation OT is not introduced when it is used the first time, but later only.

No details are provided about how the two reference methods DAT and MDAN were implemented for the experiments done in this work. When comparing the performance numbers (e.g. PESQ scores) of DAT in this paper and the numbers of DAT in its original paper [19], they are different for similar testing conditions; of course, various settings can impact on the results, which makes it important to explain the implementation details and ensure the reference works are reproduced correctly.



**Time Spent Reviewing:**

4

---

> ### Author Response · Authors · 2021-08-10
> **Reply to Reviewer Ryjo**
>
> We thank the reviewer for pointing out the clarity may be further improved by the suggestions provided.
>
> **To the Reviewer's comments:**
>
> > **Optimal transport (OT) is the theory fundamental to this work. However, it is neither well explained (thinking OT is less known for the speech enhancement community) nor well motivated (namely why it is a good solution for this problem).**
>
> - We appreciate the reviewer’s advice on more detailed motivations and explanations on OT.  Mathematically, OT concerns how different two probability distributions are (e.g., https://arxiv.org/pdf/1801.07745.pdf) and finds the best transportation plan to move one distribution to the other. Owing to the idea of transportation between two probability distributions, OT has a natural connection with the domain alignment in the context of domain adaptation. Such intuition and motivation lay the foundation for our model.
> When a revision is allowed, more explanations and motivations shall be provided in Sec. 3.2.
>
>
> > **The abbreviation OT is not introduced when it is used the first time, but later only.**
> - The abbreviation OT will be fixed to the right place when a revision to the manuscript is allowed.
>
> > **No details are provided about how the two reference methods DAT and MDAN were implemented for the experiments done in this work. When comparing the performance numbers (e.g. PESQ scores) of DAT in this paper and the numbers of DAT in its original paper [19], they are different for similar testing conditions; of course, various settings can impact on the results, which makes it important to explain the implementation details and ensure the reference works are reproduced correctly.**
>
> - It was due to the readability and the page limit that the implementation details were not placed in the main draft. However, the core information of all implementations was carefully addressed in Sec.1 of the supplementary materials submitted (e.g., node numbers, layer structures, network topology and architectures…, etc) to provide the technical construction for interested readers. The full codes of other two methods (MDAN & DAT) will be further organized in a repository tutorial and made public later when the review period ends. In particular, the DAT code can be directly downloaded from their Github to perform domain adaptation to fit our purpose.
> The deviation of the DAT scores between the original paper and ours indeed may be due to various settings: first, the source domain types did not all coincide (their noise types: *“car”, “engine”, “soft wind”, “strong wind”, and “pink”; ours: “car”, “engine”, “pink”, “wind”, “cabin”*) and even so we may not have exactly same audio on those intersecting noise types. Moreover, the training and testing utterances had different numbers and distinct random choices. In addition, we had 9 training SNRs, but they had only 6 SNRs. These factors may result in the fluctuations of DAT scores we observed. Notably, the DAT scores reported in our manuscript were performed using the optimal parameters suggested in their (official) Github.

---

### Decision · Program_Chairs · 2021-09-27

**Decision:**

Accept (Poster)

**Comment:**

This paper proposes a discriminator-constrained optimal transport network (DOTN) for unsupervised speech enhancement.  The authors apply joint distribution optimal transport with a discriminator under Wasserstein GAN to generate enhanced speech signals.  The novelty of the work is the combination of joint distribution OT and WGAN for both domain adaptation and unsupervised learning simultaneously in the application of speech enhancement, which is a regression setting.  The authors conduct experiments on Voice Bank and TIMIT datasets under various conditions and show good performance under the PESQ and STOI metrics.  The paper is well written.  The work is theoretically solid and the performance under PESQ and STOI is decent.  However, there are a few standing concerns. First of all, the authors should make it clear that both joint distribution optimal transport and WGAN are existing techniques by directly citing the previous work when mathematically formulating the problem in Section 3.2 even though they are mentioned in the related work.  Second, there has been a significant concern regarding the quality of the generated speech by DOTN in the demo where distortions are quite noticeable perceptually.  It is not clear whether this is due to the unsupervised setting.  As suggested by one of the reviewers, the authors upload the comparative performance between MetricGAN+ and the proposed DOTN in terms of PESQ and STOI.  It shows that DOTN has slightly better STOI and is quite a bit lower in PESQ since SOTA of MetricGAN+ is optimized under PESQ.  This appears to clear up the concern to some degree.  All reviewers consider the work interesting and the authors are responsive in the discussion period.  I would recommend acceptance.  But the authors should include the additional results in the rebuttal and discussion in the revised version.  Furthermore, it would be greatly helpful to include subjective evaluation results in the revision to make the work more convincing.